# GRAPH REPRESENTATION LEARNING WITH MULTI-GRANULAR SEMANTIC ENSEMBLE

## ABSTRACT

Self-supervised learning (SSL) has garnered increasing attention in the graph learning community, owing to its capability of enabling powerful models pre-trained on large unlabeled graphs for general purposes, facilitating quick adaptation to specific domains. Though promising, existing graph SSL frameworks often struggle to capture both high-level abstract features and fine-grained features simultaneously, leading to sub-optimal generalization abilities across different downstream tasks. To bridge this gap, we present *Multi-granularity Graph Semantic Ensemble via Knowledge Distillation*, namely **MGSE**, a plug-and-play graph knowledge distillation framework that can be applied to any existing graph SSL framework to enhance its performance by incorporating the concept of multi-granularity. Specifically, MGSE captures multi-granular knowledge by employing multiple student models to learn from a single teacher model, conditioned by probability distributions with different granularities. We apply it to six state-of-the-art graph SSL frameworks and evaluate their performances over multiple graph datasets across different domains, the experimental results show that MGSE can consistently boost the performance of these existing graph SSL frameworks with up to 9.2% improvement.

## 1 INTRODUCTION

Graph neural network (GNN) garnered increasing attention due to its exceptional performance in learning powerful representations over graphs (Kipf & Welling, 2017). However, obtaining a sufficient number of annotated graphs can be prohibitively expensive. As a result, self-supervised learning (SSL) on graphs has emerged as a popular research direction, offering the ability to learn task-agnostic representations without relying on costly label annotations (Zhu et al., 2020; You et al., 2020; Thakoor et al., 2022). The SSL paradigm has proven particularly valuable in various real-world applications, such as drug discovery (Gilmer et al., 2017; Wu et al., 2018; Zhang et al., 2021b), protein analysis (Jiang et al., 2017), and social network analysis (Fan et al., 2019; Wang et al., 2019a; Ying et al., 2018), where the scarcity of labeled data poses a significant challenge. Meanwhile, SSL frameworks are usually time-consuming and computationally expensive, because they require either contrasting the positive samples with a large number of negatives (Chen et al., 2020a) or learning the same object from different views by multiple trials (Chen & He, 2021), which undermines the feasibility to train a model from scratch on each application. To overcome the computational barrier, transfer learning frameworks have been extensively explored to quickly migrate models trained for general purposes to task-specific domains. For example, models can be adapted to new downstream tasks by appending only one or few MLP layers (Chen et al., 2020a; Kenton & Toutanova, 2019) with little fine-tuning to achieve good performance.

Unfortunately, the knowledge from models trained for general purposes may not always suffice for various downstream tasks requiring semantics with different granularities. For instance, in the computer vision field, object detection requires fine-grained knowledge to recognize minute objects; whereas image segmentation needs a high-level overview of the contours (Sun et al., 2019). Similarly, in the natural language processing field, named entity recognition requires fine-grained token-level representations but document classification needs high-level understanding over the whole input text (Liu et al., 2022). Consider the illustration in Figure 1, where amino acids share common amino and carboxyl groups, which make up the majority of amino acids. However, their hydrophobic properties are determined by the side chains, with glycine and alanine being hydrophobic and serine and threonine being non-hydrophobic. The defining sub-structures for amino acids

operate at a coarse granularity, while those for hydrophobicity operate at a fine-grained granularity. Hence, the incorporation of multi-granularity into graph learning could help the model gain a more comprehensive understanding of the underlying graph structure and its associated semantics.

Existing SSL methods design different tasks and objectives to enhance downstream task performance, but few of those leverage the concept of multi-granularity when applied to different tasks. For instance, graph contrastive learning methods (You et al., 2020; 2021; Suresh et al., 2021) learn from instance discrimination tasks (Tschannen et al., 2019), which maximize the similarities between the positive pairs while minimizing those between different instances, including instances falling under similar granularities. Consequently, the graph representations learned through contrastive objectives (e.g., InfoNCE (Oord et al., 2018)) might be optimized into sub-optimal situations, where high-level abstraction features are neglected and only those fine-grained features are kept. On the other hand, graph SSL methods based on generative objectives (Hou et al., 2022; Hu et al., 2020b) struggle to extract high-level

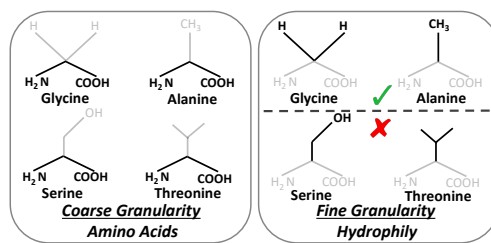

Figure 1: Multi-granularity existing in graphs. With different granularities, the activated sub-structures diverge significantly. Coarse granularity captures the high-level semantics; whereas fine granularity captures minute sub-structures. Bold sub-graphs indicate the activated sub-graphs in the current granularity. The dashed line indicates the granularity split.

semantics since the generative objectives require the model to recover the original masked attributes, enforcing the encoder to learn representations at fine-grained levels, which falls short on tasks whose objective naturally conforms the high-level abstraction features (Liu et al., 2021). GraphLoG (Xu et al., 2021) explores the prototypical graph contrastive learning, which indeed considers the concept of representation granularity. However, it suffers from the same issue because the cluster granularity is pre-defined before the model training, which is not flexible to handle all downstream tasks.

In light of this, we propose to combine different models in an ensemble manner, where each candidate model learns at a specific granularity. To learn graph representations at multiple granularities, one straightforward design could be training multiple GraphLoG models and combining them during the inference phase. However, such an implementation might entail sub-optimal performance because the hierarchical granularities of different GraphLoG models might overlap due to the independent training of each model. Instead, we explore knowledge distillation (KD), by which student models with different granularities learn from the same pre-trained teacher model, such that the knowledge learned by student models is guaranteed to be non-overlapping. KD is initially proposed for model compression (Hinton et al., 2014; Zhang et al., 2022), however, recently KD has also been proven to enhance the model performance: distilling knowledge of one model to its randomly initialized clone without extra supervision signal (i.e., self-distillation) results in performance improvement (Zhang et al., 2019b; Furlanello et al., 2018; Chen et al., 2020b; Tian et al., 2019). Previous Study (Allen-Zhu & Li, 2020) explains this phenomenon from the point of ensemble learning and theoretically proves that the student model leans to learn a perspective different from that of the teacher model due to the various random initialization for model parameters. The student model learns the representations through its own parameters while leveraging the knowledge learned by the teacher model initialized differently, which improves the performance (e.g., instances misclassified by the model of one perspective could be correctly predicted by the model of the other perspective, and vice versa). To learn the multi-granular knowledge in graphs, leveraging the philosophies and the performance improvement brought by KD, we propose *Multi-granularity Graph Semantic Ensemble via Knowledge Distillation*, namely **MGSE**, a plug-and-play knowledge distillation framework aimed at enhancing the transfer learning performance for existing graph SSL models. Unlike existing graph KD frameworks that require specific configurations, MGSE makes no such assumption and is directly applicable to any graph SSL method to improve its performance. Specifically, we extract the multi-granular knowledge by training multiple student models that align their outputs with the teacher model in the hidden space with different granularities. Finally, we combine the knowledge learned by students with different granularities in an ensemble manner to fulfill the downstream applications. Our contributions are summarized into three aspects: (i) We propose the **MGSE**, a novel plug-and-play knowledge distillation framework aimed at enhancing the generalization ability of any graph SSL model by the incorporation of multi-granularity; (ii) We provide theoretical illustration to demonstrate the superiority of our proposed framework; (iii) Comprehensive experiments are conducted and the

results show that all graph SSL models outperform their original variants when equipped with MGSE, indicating the strong performance improvement brought by introducing multi-granularity.

## 2 RELATED WORK

**Graph Neural Network.** Graph neural networks (GNNs) map the non-Euclidean graph-structured data into lower-dimensional hidden spaces for further utilization of the graph learning task. The key mechanism behind GNNs is message passing, where GNNs learn node representations by transforming and aggregating the information along the edges of graphs (Scarselli et al., 2008; Kipf & Welling, 2017). Therefore, the learned node representations are generally optimized to preserve the proximity features, i.e., the node representations reflect their neighborhood distributions (Ma et al., 2022). GNN variants mainly focus on improving the transformation or aggregation functions (Veličković et al., 2018; Xu et al., 2018; Gasteiger et al., 2019; Hamilton et al., 2017; Zhang et al., 2019a; Wang et al., 2019b), achieving better effectiveness or scalability. Due to the remarkable performance of GNNs on graph data, they have been broadly used in many applications, such as drug discovery (Gilmer et al., 2017; Wu et al., 2018; Zhang et al., 2021b), protein analysis (Jiang et al., 2017), and social network analysis (Fan et al., 2019; Wang et al., 2019a; Ying et al., 2018).

**Self-Supervised Learning on Graph.** According to the pre-training task, existing graph SSL methods can be mainly categorized into contrast-based and generation-based methods (Liu et al., 2021). Contrastive graph SSL methods learn graph representations following the InfoMax principle (Tschannen et al., 2019), where the mutual information between positive pairs is maximized while that between negative pairs is minimized (Velickovic et al., 2019; Zhu et al., 2020; Hassani & Khasahmadi, 2020; You et al., 2020; 2021). To reduce the computational cost entailed by extensive negative pairs, some recent works also propose to achieve the same effects by either specific architecture designs such as dual-encoder architecture with stop-gradient (Thakoor et al., 2022; Yang et al., 2021b), or feature regularization terms such as feature decorrelation (Zhang et al., 2021a). On the other hand, generative graph SSL models learn graph representations by recovering features of the masked nodes or the masked edges (Liu et al., 2021; You et al., 2018; Hu et al., 2020b; Kipf & Welling, 2016; Hou et al., 2022). Despite their promising performance on many downstream tasks, those SSL methods are mainly evaluated under the in-distribution setting. They do not explicitly leverage the granularities accompanied by the downstream domains, therefore leaving certain spaces in performance improvement under some scenarios (e.g., GNNs are trained over general molecular graphs but tested on graphs from other domains). In light of this, we propose MGSE, a plug-and-play module, to provide a more comprehensive view for downstream tasks with multi-granular knowledge.

**Knowledge Distillation.** Knowledge distillation (Hinton et al., 2014) is introduced as a strategy for model compression, accomplished by having a compact student model approximate the performance of a large over-parameterized teacher model, supervised by both the annotated labels and the teacher model output. However, recently KD has also been proven to enhance the model performance: distilling knowledge of one model to its randomly initialized clone without extra supervision signal (i.e., self-distillation) leads to performance improvement(Zhang et al., 2019b; Furlanello et al., 2018; Chen et al., 2020b; Fang et al., 2020). Some (Allen-Zhu & Li, 2020) credit this to ensemble learning (Hansen & Salamon, 1990), by theoretically proving that random initialization allows the teacher and student models to learn features from different views. Assisted by the knowledge distillation strategy, the student model is able to integrate the information from its own view with the "dark knowledge" (Furlanello et al., 2018; Tian et al., 2019; Allen-Zhu & Li, 2020) learned by the teacher model from another view, therefore improving the performance While most of the existing works focus on model compression by incorporating a lighter student model (Yang et al., 2021a; Zhang et al., 2022), few of them exploit the advantage of feature ensemble brought by knowledge distillation for further model performance generalizability. In this work, we aim to exploit the merits of ensemble learning along with KD to further fortify the generalizability of existing graph SSL methods.

## 3 PRELIMINARIES

**Notations.** We use $G = (V, E)$ to represent the input graph, where $V$ and $E$ stand for the node set and edge set, respectively. Besides, we use $\mathbf{x}_v \in \mathbb{R}^{D_v}$ and $\mathbf{x}_e \in \mathbb{R}^{D_e}$ to denote the attribute vector of each node $v \in V$ and edge $e \in E$. In this work, we focus on the graph-level tasks, whose datasets

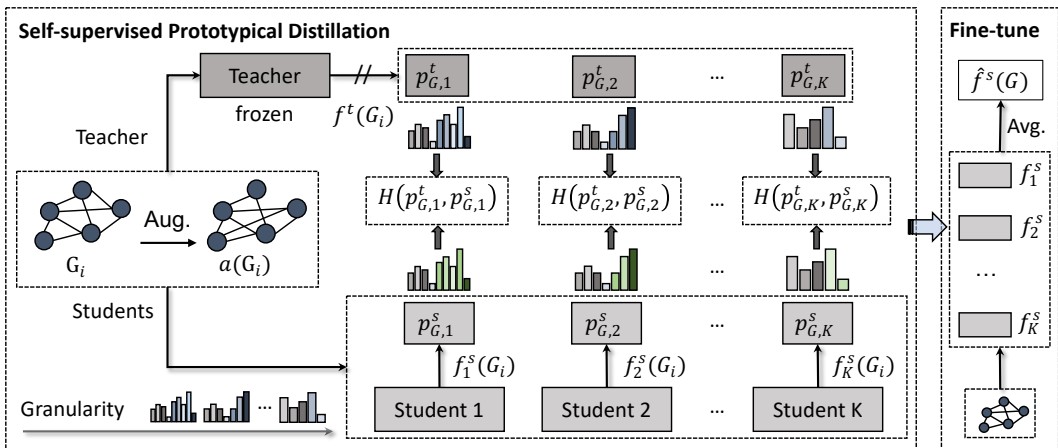

Figure 2: The illustration of our proposed MGSE. The teacher model $f^t$ can be any existing graph SSL method with its parameters frozen. $K$ randomly initialized student models are constructed with identical architecture, each with a set of prototypes initialized to transfer the semantic information from the teacher in a specific granularity. During the distillation, the outputs of the teacher and the student are aligned at different granularities. After that, all students are independently fine-tuned and the average of their predictions are taken as the final prediction.

include multiple graphs, i.e., $\mathcal{G} = \{G_i = (V_i, E_i)\}_{i=1}^N$ and $c$ is the number of classes. Each graph $G_i$ is associated with an $C$-dimensional one-hot prediction target vector, denoted as $\mathbf{y}_i \in \mathbb{R}^C$.

**Graph Neural Networks.** As we mentioned in Section 2, most GNNs follow the message-passing schema to iteratively update the node embedding through the message from its neighborhood $\mathcal{N}_v$. Considering a $L$-layer GNN model, the forward process of the $l$-th layer can be formulated as:

$$\mathbf{h}_v^{(l+1)} = \text{COM}\left(\mathbf{h}_v^{(l)}, \text{AGG}\left(\left\{\mathbf{h}_u^{(l)}, \forall u \in \mathcal{N}_v\right\}\right)\right), \tag{1}$$

where $\mathbf{h}_v^{(l)}$ is the embedding of node $v$ at the $l$-th layer and $\mathbf{h}_v^{(0)} = \mathbf{x}_v$, $\mathcal{N}_v$ is the set of the nodes adjacent to $v$, AGG and COM are aggregation function and transformation function of the GNN layer, respectively. For the graph classification task, there is a READOUT function (e.g., mean, sum, or max) to summarize all the node embeddings into one graph embedding after $L$-th layer.

## 4 METHODOLOGY

In this section, we will introduce the technical details of our proposed Multi-granular Graph Ensemble via Knowledge Distillation (MGSE), whose overall framework is shown in Figure 2. MGSE enhances the representations learned from student models by the self-supervised knowledge distillation, such that the resulting representations achieve better generalization ability. Given the input graph $G$ and a pre-trained teacher model $f^t$ (including a GNN encoder $f_g$ and a projection layer $f_h$), we transfer the knowledge from $f^t$ to the student model $f^s$ by aligning their hidden representations. Multi-granular semantic knowledge is extracted by training multiple student models with different granularities. At last, we unify the distilled student models through model ensemble to conduct downstream inference.

### 4.1 SELF-SUPERVISED PROTOTYPICAL DISTILLATION

Similar to the previous self-supervised distillation work (Fang et al., 2020), we keep the pre-trained teacher model frozen while training the randomly initialized student models to approximate its output. However, the semantic granularity is challenging to extract in the SSL setting due to the absence of supervision. Inspired by prototypical contrastive learning frameworks that encode features at one granular level (Li et al., 2021; Guo et al., 2022; Xu et al., 2021), we align the outputs of the teacher model and the student model in various probability spaces to extract features in multiple granularities.

Given a graph $G_i \in \mathcal{G}$ and its augmented view $a(G_i)$, where $a(\cdot)$ is randomly sampled from a graph augmentation family $\mathcal{A}$ (i.e., node dropping, edge perturbation and subgraph sampling), we firstly get the representations of these two graphs from the pre-trained teacher model, i.e., $\mathbf{z}_i^t = f^t(G_i) \in \mathbb{R}^D$. Next, we randomly initialize $K$ sets of corresponding trainable prototypes to map $\mathbf{z}_i^t$ into $K$ probability

spaces. The prototype matrix for $k$-th set is denoted as $\mathbf{q}_k \in \mathbb{R}^{D \times D_k}$, where each column indicates the centroid of a prototype and $D_k$ is the number of prototypes in $k$-th set. In order to extract multi-granular semantics from a single teacher model $f^t$, we thereby train multiple student models specific to different granularities. Given $K$ sets of prototypes corresponding to different granular semantics, we train $K$ student models with identical architecture, i.e., $\left\{\mathbf{z}_{i,k}^s\right\}_{k=1}^K = \{f_k^s(G_i)\}_{k=1}^K$, to obtain the approximated probability distributions of the teacher model in different granularities. The soft prototypical assignment for $\mathbf{z}_i^t$ and $\mathbf{z}_{i,k}^s$ in $k$-th prototype set are denoted as $\mathbf{p}_{i,k}^t$ and $\mathbf{p}_{i,k}^s$, with $d$-th dimension calculated as:

$$\left[\mathbf{p}_{i,k}^t\right]_d = \frac{\exp\left(\bar{\mathbf{z}}_i^t \cdot [\bar{\mathbf{q}}_k]_d / \tau_t\right)}{\sum_{d'=1}^{D_k} \exp\left(\bar{\mathbf{z}}_i^t \cdot [\bar{\mathbf{q}}_k]_{d'} / \tau_t\right)}, \quad \left[\mathbf{p}_{i,k}^s\right]_d = \frac{\exp\left(\bar{\mathbf{z}}_{i,k}^s \cdot [\bar{\mathbf{q}}_k]_d / \tau_s\right)}{\sum_{d'=1}^{D_k} \exp\left(\bar{\mathbf{z}}_{i,k}^s \cdot [\bar{\mathbf{q}}_k]_{d'} / \tau_s\right)}, \tag{2}$$

where $\tau_t, \tau_s \in (0,1)$ is the temperature parameter for the teacher and students, $\bar{\mathbf{z}}_i^t$, $\bar{\mathbf{z}}_{i,k}^s$ and $\bar{\mathbf{q}}_k$ is the $L_2$ normalized representations and prototypes and $[\cdot]_d$ is the $d$-th element in a variable. To guarantee different prototype sets capture different semantic granularities, we assign the number of prototypes in each set in descending order, i.e., $D_1 > D_2 > \cdots > D_K > 1$. In short, we get $K$ sets of "soft label" by measuring the cosine similarity between $\mathbf{z}_i^t$ and centroids of different prototype sets, then the soft labels are utilized as supervision to align the output of teacher and students. It is noteworthy that we use a larger temperature parameter $\tau_s$ than $\tau_t$ to implicitly encourage the student models to produce confident low entropy anchor predictions (Fang et al., 2020; Caron et al., 2021). In our implementation, we use a sharpen operation (i.e., $\tau_t = \tau_s^{1/P}$ with $P \in \mathbb{R}^+$ and $P < 1$) on the target probability distribution produced by the teacher model to achieve the same effect. With the multi-granular distributions from the student models and their corresponding distributions from the teacher model, MGSE is optimized by minimizing the divergence between the probability distributions of the teacher and the students. We use the negative cross entropy to measure the divergence for each granularity and for $k$-th granularity:

$$\mathcal{D}\left(\mathbf{p}_k^t, \mathbf{p}_k^s\right) = -\mathcal{H}\left(\mathbf{p}_k^t, \mathbf{p}_k^s\right) = \frac{1}{ND_k} \sum_{i=1}^N \sum_{d=1}^{D_k} - \left[\mathbf{p}_{i,k}^t\right]_d \cdot \log\left[\mathbf{p}_{i,k}^s\right]_d. \tag{3}$$

Note that the distillation objective in Equation 3 can maximize the mutual information between the output of the teacher model and student model, which is similar to InfoMax principle (Tschannen et al., 2019) in contrastive SSL frameworks. Particularly, our optimization objective can be an analogy to the ProtoNCE loss to maximize the mutual information at cluster-level. We conduct a more detailed discussion in Appendix A. Therefore, by aligning the representation probability distributions of the teacher model and the student models, we can transfer the knowledge from the teacher to students.

## 4.2 TRAINING OBJECTIVE

To train the student models, we use the cross entropy measurement $H\left(\mathbf{p}_{i,k}^t, \mathbf{p}_{i,k}^s\right)$ to match the probability distributions of the teacher model and student models in different granularities. Besides, a mean entropy maximization (ME-MAX) (Assran et al., 2021) regularization term is explored to maximize the entropy of outputs of student models so that all prototypes are enforced to be utilized, thereby the learned representations can be more semantically discriminative. Similar to Equation 3, the regularization term is formulated as:

$$\mathcal{H}(\bar{\mathbf{p}}_k^s) = \left[\frac{1}{N} \sum_{i=1}^N \mathbf{p}_{i,k}^s\right] \cdot \log\left[\frac{1}{N} \sum_{i=1}^N \mathbf{p}_{i,k}^s\right]. \tag{4}$$

Therefore, we can get the overall optimization objective of our proposed MGSE below:

$$\mathcal{L} = \frac{1}{KN} \sum_{k=1}^K \sum_{i=1}^N \left[\mathcal{D}\left(\mathbf{p}_{i,k}^t, \mathbf{p}_{i,k}^s\right) - \lambda \mathcal{H}(\bar{\mathbf{p}}_k^s)\right], \tag{5}$$

where $\lambda > 0$ is the weight of the ME-MAX regularizer. During the training process, the $K$ sets of prototypes and student models will be jointly optimized using the objective function in Equation

5 via the same optimizer. By combining the distribution sharpening operation and the ME-MAX regularization, the $D_k$ prototypes within each set are optimized to be distinctive, capturing distinct semantic features at specific granularities. Consequently, the extraction of semantic features at various granularities is facilitated by aggregating prototype sets with different $D_k$ values. We make a further theoretical discussion to explain the reason in Proposition 1, and the proof is provided in Appendix B.

**Proposition 1.** *Given the target probability distribution $\mathbf{p}_k^t$ produced by the teacher model and the anchor probability distribution $\mathbf{p}_k^t$ produced by the student model, $\|\nabla\mathcal{D}\left(\mathbf{p}_k^t, \mathbf{p}_k^s\right)\| + \|\nabla\mathcal{H}(\bar{\mathbf{p}}_k^s)\| > 0$ if there exist representation collapse, i.e., $\mathbf{p}_{i,k}^t = \mathbf{p}_{j,k}^s = \frac{1}{K}\mathbf{1}_K$ for $\forall G_i, G_j \in \mathcal{G}$.*

### 4.3 MULTI-GRANULAR MODEL ENSEMBLE

To incorporate the knowledge from multiple granularities, we thus combine multiple distilled student models in an ensemble manner to fully utilize the multi-granular knowledge. During the finetuning process, we update $K$ student models from the multi-granular self-distillation process to adapt to the downstream tasks. It is noteworthy that the $K$ student models use the same training set as the teacher model does. We produce the final prediction by averaging the predictions of $K$ student models as:

$$\hat{f}^s(G) = \sum_{k=1}^{K} f_k^s\left(G\right)/K. \tag{6}$$

We derive Theorem 1 to justify the ensemble of the multi-granular student models can improve the prediction accuracy and robustness compared with the single model, shown as the following:

**Theorem 1.** *For a graph dataset $\mathcal{G}$ and multiple student model $\{f_k^s\}_{k=1}^K$ with the different initialized knowledge, the ensemble of their predictions can reduce the error rate compared with the prediction of any single model, i.e,*

$$\mathbb{E}_{\mathcal{G}}\left[\mathbf{1}\left(\hat{f}^s(G) = y\right)\right] \geq \mathbb{E}_{\mathcal{G}}\left[\mathbf{1}\left(f_k^s(G) = y\right)\right] \quad \forall k \in [K],$$

where $y$ is the ground truth. More detailed derivation for this theorem is provided in Appendix C. Besides, the computation cost of our proposed MGSE only grows linearly w.r.t. the number of student models. Given the computation cost of existing graph SSL methods as $O(N + M)$, where $N$ is the number of nodes and $M$ is the number of edges, the computation cost of our proposed MGSE will be $O(K(N + M))$. We provide more empirical analysis of the model depth and student numbers to demonstrate the scalability of our method in the experiment section.

## 5 EXPERIMENTS

In this section, we demonstrate the performance improvement brought by our proposed MGSE after being applied to state-of-the-art graph SSL frameworks. Extensive experiments are conducted on public graph benchmark datasets in different domains to test the effectiveness of the learned representation. Besides, we provide more experiments to analyze the designs of our proposed framework. Due to the page limitation, we only demonstrate the empirical study on part of the datasets in the experiments section. Other contents, including implementation details, experimental results on node-level tasks and more empirical analysis are provided in Appendix E and F.

### 5.1 EXPERIMENT SETUP

**Datasets.** For the graph-level tasks, we follow the setting adopted by previous works (You et al., 2020; Suresh et al., 2021) to evaluate our proposed MGSE with the data from the chemistry domain and biology domain under the transfer learning setting. For the chemistry domain datasets, the teacher model and student models are pre-trained on the ZINC15 dataset (Sterling & Irwin, 2015) which contains 2 million unlabeled molecule graphs. And then we evaluate the distilled student models on eight downstream datasets with binary or multi-class classification tasks. Both the pre-train dataset and the fine-tuning datasets are extracted from MoleculeNet (Wu et al., 2018) and are split by the scaffold to simulate the real-world scenario where the concept of multi-granularity improves the performance significantly. The pre-train dataset of the biology domain is PPI-306K (Hu et al., 2019) and the downstream application is to predict biological functions of 88K protein ego-networks,

Table 1: Performance (i.e., AUC) of state-of-the-art SSL-based GNN frameworks in the transfer learning setting, and improvements after MGSE is applied. The percentage in the parentheses refers to the percentage of performance improvement brought by MGSE.

| Model | BBBP | Tox 21 | ToxCast | SIDER | ClinTox | HIV | BACE | MUV |
|---|---|---|---|---|---|---|---|---|
| No Pre-train | $65.8_{\pm 4.5}$ | $74.0_{\pm 0.8}$ | $63.4_{\pm 0.6}$ | $57.3_{\pm 1.6}$ | $58.0_{\pm 4.4}$ | $75.3_{\pm 1.9}$ | $70.1_{\pm 5.4}$ | $71.8_{\pm 2.5}$ |
| GraphCL | $69.68_{\pm 0.67}$ | $73.87_{\pm 0.66}$ | $62.40_{\pm 0.57}$ | $60.53_{\pm 0.88}$ | $75.99_{\pm 2.65}$ | $78.47_{\pm 1.22}$ | $75.38_{\pm 1.44}$ | $69.80_{\pm 2.66}$ |
| +MGSE | $72.26_{\pm 0.65}$ | $75.89_{\pm 0.33}$ | $64.57_{\pm 0.34}$ | $61.44_{\pm 0.68}$ | $78.67_{\pm 2.89}$ | $79.07_{\pm 0.72}$ | $79.22_{\pm 0.93}$ | $71.46_{\pm 1.45}$ |
| Perf. (↑) | +2.58 (3.7%) | +2.02 (2.7%) | +2.17 (3.5%) | +0.91 (1.5%) | +2.68 (3.5%) | +0.60 (0.8%) | +3.84 (5.1%) | +1.66 (2.4%) |
| RGCL | $71.42_{\pm 0.66}$ | $75.20_{\pm 0.34}$ | $63.33_{\pm 0.17}$ | $61.38_{\pm 0.61}$ | $83.38_{\pm 0.90}$ | $77.90_{\pm 0.80}$ | $76.03_{\pm 0.77}$ | $76.66_{\pm 0.99}$ |
| +MGSE | $71.65_{\pm 0.78}$ | $76.82_{\pm 0.62}$ | $64.85_{\pm 0.20}$ | $63.72_{\pm 0.63}$ | $84.88_{\pm 2.01}$ | $78.33_{\pm 0.85}$ | $77.40_{\pm 1.27}$ | $77.18_{\pm 0.81}$ |
| Perf. (↑) | +0.23 (0.3%) | +1.62 (2.2%) | +1.52 (2.4%) | +2.34 (3.8%) | +1.50 (1.8%) | +0.43 (0.6%) | +1.37 (1.8%) | +0.52 (0.7%) |
| AD-GCL | $70.00_{\pm 1.07}$ | $76.54_{\pm 0.82}$ | $63.07_{\pm 0.72}$ | $63.28_{\pm 0.79}$ | $79.78_{\pm 3.52}$ | $78.28_{\pm 0.97}$ | $78.51_{\pm 0.80}$ | $72.30_{\pm 1.61}$ |
| +MGSE | $70.44_{\pm 0.70}$ | $76.80_{\pm 0.80}$ | $64.60_{\pm 0.59}$ | $63.50_{\pm 0.92}$ | $83.05_{\pm 2.64}$ | $78.91_{\pm 0.57}$ | $79.65_{\pm 1.07}$ | $74.32_{\pm 0.85}$ |
| Perf. (↑) | +0.44 (0.6%) | +0.26 (0.3%) | +1.53 (2.4%) | +0.22 (0.3%) | +3.27 (4.1%) | +0.63 (0.8%) | +1.14 (1.5%) | +2.02 (2.8%) |
| JOAO | $70.22_{\pm 0.98}$ | $74.98_{\pm 0.29}$ | $62.94_{\pm 0.48}$ | $59.97_{\pm 0.79}$ | $81.32_{\pm 2.49}$ | $76.73_{\pm 1.23}$ | $77.34_{\pm 0.48}$ | $71.66_{\pm 1.43}$ |
| +MGSE | $71.93_{\pm 0.50}$ | $76.20_{\pm 0.33}$ | $64.26_{\pm 0.27}$ | $61.02_{\pm 0.86}$ | $83.30_{\pm 2.44}$ | $77.50_{\pm 0.67}$ | $79.82_{\pm 0.71}$ | $73.52_{\pm 0.62}$ |
| Perf. (↑) | +1.71 (2.4%) | 1.22 (1.6%) | +1.32 (2.1%) | +1.05 (1.8%) | +1.98 (2.4%) | +0.77 (1.0%) | +2.48 (3.2%) | +1.86 (2.6%) |
| GraphMAE | $72.0_{\pm 0.6}$ | $75.5_{\pm 0.6}$ | $64.1_{\pm 0.3}$ | $60.3_{\pm 1.1}$ | $82.3_{\pm 1.2}$ | $77.20_{\pm 1.0}$ | $83.1_{\pm 0.9}$ | $76.3_{\pm 2.4}$ |
| +MGSE | $71.62_{\pm 0.51}$ | $76.52_{\pm 0.48}$ | $65.31_{\pm 0.38}$ | $62.46_{\pm 0.52}$ | $84.41_{\pm 2.20}$ | $78.03_{\pm 0.70}$ | $82.92_{\pm 0.75}$ | $77.15_{\pm 0.75}$ |
| Perf. (↑) | -0.38 (-0.5%) | +1.02 (1.4%) | +1.21 (1.9%) | +2.16 (3.6%) | +2.11 (2.6%) | +0.83 (1.1%) | -0.18 (-0.2%) | +0.85 (1.1%) |
| GraphLoG | $72.5_{\pm 0.8}$ | $75.7_{\pm 0.5}$ | $63.5_{\pm 0.7}$ | $61.2_{\pm 1.1}$ | $76.7_{\pm 3.3}$ | $77.8_{\pm 0.8}$ | $83.5_{\pm 1.2}$ | $76.0_{\pm 1.1}$ |
| +MGSE | $72.57_{\pm 1.13}$ | $76.84_{\pm 0.58}$ | $64.88_{\pm 0.39}$ | $63.08_{\pm 0.86}$ | $83.72_{\pm 2.02}$ | $78.64_{\pm 0.80}$ | $83.18_{\pm 1.24}$ | $77.22_{\pm 0.94}$ |
| Perf. (↑) | +0.07 (0.1%) | +1.14 (1.5%) | +1.38 (2.2%) | +1.88 (3.1%) | +7.02 (9.2%) | +0.84 (1.1%) | -0.32 (-0.4%) | +1.22 (1.6%) |
| Avg. Perf. (↑) | +0.78 (1.1%) | +1.21 (1.6%) | +1.52 (2.4%) | +1.43 (2.3%) | +3.09 (3.9%) | +0.68 (0.9%) | +1.39 (1.8%) | +1.36 (1.9%) |

where the downstream dataset is split according to the species. For the node-level tasks, we follow the SSL evaluation setting of the BGRL and GraphMAE to evaluate our proposed model in Cora, Pubmed (Yang et al., 2016) and ogbn-arxiv (Hu et al., 2020a). More detailed datasets statistics are demonstrated in Appendix D.

**Baselines.** To fully demonstrate the capability of our proposed MGSE in improving the performance of existing graph SSL paradigms under the transfer learning setting, we implement multiple state-of-the-art contrastive and generative graph SSL models as the teacher models, including GraphCL (You et al., 2020), JOAO (You et al., 2021), AD-GCl (Suresh et al., 2021), GraphLoG (Xu et al., 2021), RGCL (Li et al., 2022), GraphMAE (Hou et al., 2022) and BGRL (Thakoor et al., 2022). We explore the same setting as recommended by the authors.

**Evaluation Protocol.** We follow the settings adopted by previous works (You et al., 2020; 2021) and adopt ROC-AUC as the evaluation metrics of graph-level tasks. Each experiment is repeated 10 times with a different random initialization. In the meantime, we follow the previous works to (Hou et al., 2022; Thakoor et al., 2022) to use the accuracy score as the evaluation metrics of the node-level tasks and repeat each experiment 5 times. We report both the mean metric score and its standard deviations as the final evaluation results.

## 5.2 RESULTS ANALYSIS

**Overall Performance Comparison in Graph-level Tasks.** In Table 1, we report the performance of the six teacher models and their corresponding multi-granular graph semantic ensemble trained by our proposed MGSE framework on the eight chemistry domain datasets. Specifically, we also add the experimental results without SSL (denoted as "No pre-train"), where a randomly initialized 5-layer-GIN model is trained from scratch to conduct inference on the eight datasets. From the table we can reach three observations: (1) Our proposed MGSE can consistently boost the performance of existing graph SSL models by an obvious margin, which demonstrates the effectiveness of learning multi-granular representations under the transfer learning setting. The dataset shift between the pre-train phase and fine-tuning phase causes a divergence between the knowledge learned from pre-train tasks and the knowledge useful for downstream tasks. This issue can be further deteriorated by the split strategy (i.e., splitting according to the scaffold) which mimics an out-of-distribution setting. In this case, low-level fine-grained features are not general enough to adapt to various tasks and consistently achieve good performances. On the contrary, the multi-granular representations can abstract the data distribution in multiple hierarchies and granularities, thereby being less affected by the out-of-distribution setting. This conclusion is also supported by the superior performance of GraphLoG over most of the other baselines, which is based on the prototypical contrastive learning method. (2) The performance improvement brought by our proposed MGSE is more significant on the multi-label classification problem, including Tox21, ToxCast, SIDER, ClinTox, and MUV. The relative performance improvements on these five datasets are 1.6% ∼ 3.9% and 0.9% ∼ 1.8% on

Table 2: Performance of the four state-of-the-art SSL frameworks on the biology domain dataset, where "*" stands for our reproduced results due to the different experimental settings employed in the original works.

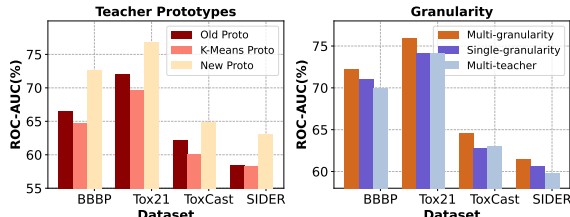

Figure 3: Impacts of different prototypes initialization strategies (left), and single-granularity versus multi-granularity (right).

| Dataset | GraphCL | AD-GCL | JOAO | GraphLoG* |
|---|---|---|---|---|
| PPI | $67.88_{\pm0.85}$ | $68.83_{\pm1.26}$ | $64.43_{\pm1.38}$ | $66.95_{\pm1.32}$ |
| | +MGSE | | | |
| | $69.11_{\pm0.70}$ | $68.95_{\pm0.83}$ | $65.37_{\pm0.96}$ | $68.26_{\pm1.06}$ |
| Perf. (↑) | 1.23 (1.8%) | 0.12 (0.2%) | 0.96 (1.5%) | 1.31 (2.0%) |

the other three datasets with single-label. This observation can be used to support our claim that it is necessary to utilize multi-granular semantic information to analyze and explain the property of a graph. A graph might own multiple properties and they could be determined by either coarse features or fine-grained features. Therefore it is more comprehensive to explain a graph by multiple granularities from coarse to fine. (3) The performance improvements over the baselines with trainable augmentation operation on the graph structure, including AD-GCL and RGCL, are relatively limited compared to the others. Those baselines implement strong regularization on the graph topology information by deleting nodes or edges to push the optimized representations focusing on the most salient and coarse features. Such a strategy could improve the performance robustness to some extent, especially when the data distribution shift is not obvious. However, it may not suit the multi-label classification scenario where we need the features in different aspects and granularities. Besides, we demonstrate the experimental results on the biology domain dataset in Table 2, from which we can find similar observations as we analyzed above. Our knowledge distillation framework can still consistently boost the performance of the adopted teacher model, and the performance improvements are more obvious for the teacher model without strong augmentation. More experimental results of the node-level tasks can are provided in Appendix F.

**Importance of Random Prototype Initialization Strategy.** In our proposed MGSE, we randomly initialize the prototypes and train them for multi-granular representation clustering. Besides the random initialization strategy, we can also run the K-means algorithm based on the output of the teacher model to compute the cluster centroids and initialize them as prototypes, denoted as **K-Means Proto**. For the prototypical graph self-supervised teacher model, i.e., GraphLoG, we can also reuse the pre-trained prototypes for distillation, denoted as **Old Proto**. Specifically, the parameter of pre-trained prototypes will be frozen together with the teacher model. We demonstrate the empirical result of the two prototype strategies and MGSE (**New Proto**) in the left subplot of Figure 3, from which we can find that the performance is much better when we use the new prototypes for the teacher model. A possible explanation is that the pre-trained prototypes could prohibit the student models from learning features in different granularities, and thus the distilled student models can not outperform or even perform on par with the teacher model.

**Importance of Multi-granularity Prototypes Setting.** To demonstrate the improvement of our methods gained from multi-granularity features (different student models are trained with prototypes in different granularities), we add two variants for comparison: (1) **Single-granularity** with only single granularity ($D_k = 50$) but the same student number, where each student is independently initialized; (2) **Multi-teacher** with different dropout ratios (0.2, 0.5, 0.8) during the fine-tuning procedure and average their output for the final prediction. In this experiment, we take GraphCL as the teacher model, the experimental results of the two variants and our proposed MGSE (**Multi-granularity**) are shown in the right subplot of Figure 3. Though the two variants based on model ensemble can also boost the teacher model's performance to some extent, the improvement over MGSE is obviously larger, indicating the effectiveness of considering multiple granularities other than the model ensemble. Meanwhile, it is fair to say that the multi-granularity semantics can not be obtained by directly setting different dropout ratios for the teacher model.

**How Prototype Regularization Enable Multi-granularity Semantics Extraction?** In the proposed MGSE, we add the ME-MAX regularization and apply the sharpening operation on the target probability distribution to avoid trivial collapsing of the trainable prototype sets. To evaluate the effectiveness of our design, we perform ablation studies by creating two model variants: (1) **w/o. sharpen**, the target probability distribution is not sharpen; (2) **w/o. ME-MAX**, the regularization weight $\lambda$ is set as 0.0. We show the experimental results in Table 3, from which we can find omitting either one of the components could fail to boost the performance of the teacher model. During distillation, the trainable prototypes are responsible for mapping the instances with various semantics

Table 3: Ablation study of the proposed MGSE with three teacher models

| Dataset | BBBP | Tox21 | ToxCast | SIDER |
|---|---|---|---|---|
| w/o. sharpen | $70.91_{\pm0.89}$ | $74.26_{\pm0.65}$ | $63.49_{\pm0.45}$ | $60.21_{\pm0.70}$ |
| w/o. ME-MAX | $70.45_{\pm1.04}$ | $73.85_{\pm0.74}$ | $63.10_{\pm0.37}$ | $60.08_{\pm0.82}$ |
| GraphCL+GMSE | $\mathbf{72.26}_{\pm0.65}$ | $\mathbf{75.89}_{\pm0.33}$ | $\mathbf{64.57}_{\pm0.34}$ | $\mathbf{61.44}_{\pm0.68}$ |
| w/o. sharpen | $69.37_{\pm0.87}$ | $74.17_{\pm0.52}$ | $64.00_{\pm0.41}$ | $59.53_{\pm1.01}$ |
| w/o. ME-MAX | $68.88_{\pm0.59}$ | $73.68_{\pm0.60}$ | $63.62_{\pm0.38}$ | $59.48_{\pm0.90}$ |
| JOAO+GMSE | $\mathbf{71.93}_{\pm0.50}$ | $\mathbf{76.20}_{\pm0.33}$ | $\mathbf{64.26}_{\pm0.27}$ | $\mathbf{61.02}_{\pm0.86}$ |
| w/o. sharpen | $70.58_{\pm0.86}$ | $74.85_{\pm0.79}$ | $64.10_{\pm0.54}$ | $61.16_{\pm1.28}$ |
| w/o. ME-MAX | $71.05_{\pm0.64}$ | $74.24_{\pm0.66}$ | $63.76_{\pm0.60}$ | $60.51_{\pm1.44}$ |
| GraphLoG+GMSE | $\mathbf{72.57}_{\pm1.13}$ | $\mathbf{76.84}_{\pm0.58}$ | $\mathbf{64.88}_{\pm0.39}$ | $\mathbf{63.08}_{\pm0.86}$ |

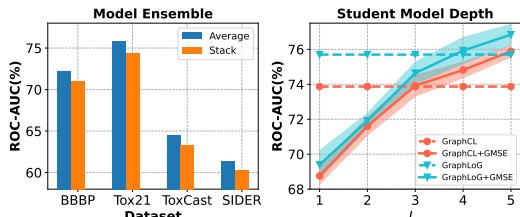

Figure 4: Impacts of different model ensemble strategies (left) and student model depth $L_s$ (right).

into different probability distributions (clustering). Without the sharpen operation, it is usually the case the student model tends to map graph instances into probability distributions with low confidence. Consequently, many instances with different semantics fall into indiscriminative representations. Meanwhile, by enforcing the full set of prototypes utilized with ME-MAX regularization, the target prototypes will be optimized to learn more discriminative features to push instances with different semantics away from each other in hidden space, thereby avoiding trivial collapsing.

**Importance of Average Model Ensemble Strategy.** Besides averaging the prediction of multiple student models for evaluation, we also conduct experiments by the stacking ensemble strategy, where we concatenate the outputs of all the student models and append a trainable linear transformation layer. The comparison results based on GraphCL are shown in the left subplot of Figure 4. We can observe that the averaging ensemble can generally outperform the stacking ensemble strategy on the four datasets. Thus we believe that for some tasks and datasets, essential features for the training graphs differ from those for the testing graphs, and hence the prediction computed from multiple independently optimized models tends to cause more robust performance.

**Potential of Reducing Computation Cost.** We conduct experiments to explore the potential of MGSE to reduce computation costs by using lighter student model architectures and smaller student numbers. Though not the research focus of this work, model compression is a byproduct of MGSE based on knowledge distillation. We vary the layer number of each student model from 1 to 5 and report the results on Tox21 dataset in the right part of Figure 4, similar observations can be found in other datasets. We can see larger model architecture always has a positive effect on the final performance, and our framework can approximate or even surpass the performance of the over-parameterized teacher model with multiple lighter student models. Thus, the results demonstrate that the extra computation cost brought by incorporating multiple student models can be alleviated by the comparable results with a lighter student model architecture.

We also vary the student number $K$ to evaluate its effects on different datasets and the experimental results are shown in Figure 5. We can see that multi-label datasets (ToxCast in the figure) tend to gain more benefits from more student models (more semantic granularities), while such diversified semantics could be unnecessary for single-label datasets (BBBP). Hence, it is reasonable to assert that our proposed method incurs a lower computational cost when applied to datasets characterized by simpler semantics. This observation additionally bolsters our argument that a wider spectrum of semantic granularities holds greater promise in effectively addressing various downstream tasks.

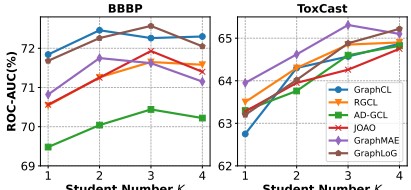

Figure 5: Impacts of different student numbers over BBBP and ToxCast.

## 6  CONCLUSION

In this paper, we study the problem of learning multi-granular representations for graphs. Inspired by the capabilities of knowledge distillation on extracting well-encoded and structural graph semantics, we propose *Multi-granularity Graph Semantic Ensemble via Knowledge Distillation*, namely **MGSE**, a plug-and-play graph knowledge distillation framework that can be easily applied to any existing graph SSL framework to enhance its performance by incorporating the concept of multi-granularity. We theoretically analyze the reason why our framework design can effectively transfer knowledge from the teacher model and further enhance its performance. Besides, we conduct extensive experiments on graph datasets across different domains, where we apply MGSE to different state-of-the-art graph SSL methods. Experimental results prove that MGSE can consistently improve the performance of all SSL methods by incorporating the concept of multi-granularity by large improvement margins up to 9.2%. More discussions on the limitations and potential impacts are provided in Appendix H.

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

# A  DISCUSSION ON THE RELATIONSHIP WITH CONTRASTIVE LEARNING

In the case when $\tau_t \to 0$, the target probability distribution $\mathbf{p}_k^t$ will approach to one-hot vector, and the objective in Equation (3) can be regarded as the ProtoNCE loss to maximize the mutual information between the output of teacher model and student model in cluster-level. And both ProtoNCE and our objective can be divided into aligning term and uniformity term. For ProtoNCE loss, we have:

$$\mathcal{L}_{ProtoNCE} = \sum_{i=1}^{N} -\log \frac{\exp\left(\mathbf{z}_i \cdot \mathbf{q}_i / \tau\right)}{\sum_{\mathbf{q}_j \sim \mathbf{q}} \exp\left(\mathbf{z}_i \cdot \mathbf{q}_j / \tau\right)} = \sum_{i}^{N} [\underbrace{-\mathbf{z}_i \cdot \mathbf{q}_i / \tau}_{\text{alignment}} + \underbrace{\log \sum_{\mathbf{q}_j \sim \mathbf{q}} \exp\left(\mathbf{z}_i \cdot \mathbf{q}_j / \tau\right)}_{\text{uniformity}}]. \tag{7}$$

And our objective function achieves similar effect:

$$\mathcal{D}\left(\mathbf{p}^t, \mathbf{p}^s\right) = \frac{1}{N} \sum_{i}^{N} \text{softmax}\left(\frac{\mathbf{z}_i^t \cdot \mathbf{q}}{\tau_t}\right) \cdot \text{softmax}\left(\frac{\mathbf{z}_i^s \cdot \mathbf{q}}{\tau_s}\right)$$

$$= \frac{1}{N} \sum_{i}^{N} \mathbf{w}_i^t \cdot [\underbrace{-\mathbf{z}_i \cdot \mathbf{q}_i / \tau_s}_{\text{alignment}} + \underbrace{\log \sum_{\mathbf{q}_j \sim \mathbf{q}} \exp\left(\mathbf{z}_i \cdot \mathbf{q}_j / \tau_s\right)}_{\text{uniformity}}], \tag{8}$$

where $\mathbf{w}_i^t$ is the soft label produced by the teacher model, we can regard it as the weight of the loss term since the teacher model is frozen.

# B  PROOF OF PROPOSITION 1

**PROPOSITION 1.** *Given the sharpened target probability distribution $\mathbf{p}_k^t$ produced by the teacher model and the anchor probability distribution $\mathbf{p}_k^t$ produced by the student model, $\|\nabla \mathcal{D}\left(\mathbf{p}_k^t, \mathbf{p}_k^s\right)\| + \|\nabla \mathcal{H}(\bar{\mathbf{p}}_k^s)\| > 0$ if there exist representation collaps, i.e., $\mathbf{p}_{i,k}^t = \mathbf{p}_{j,k}^s$ for $\forall G_i, G_j \in \mathcal{G}$.*

*Proof.* As we introduced, the normalized probability distribution $\mathbf{p}_{i,k}^t \in \mathbb{R}^{D_k}$ of teacher model in the $k$-th granularity can be computed by:

$$\left[\mathbf{p}_{i,k}^t\right]_d = \frac{\exp\left(\bar{\mathbf{z}}_{i,k}^t \cdot [\bar{\mathbf{q}}_k]_d / \tau_t\right)}{\sum_{d'=1}^{D_k} \exp\left(\bar{\mathbf{z}}_{i,k}^t \cdot [\bar{\mathbf{q}}_k]_{d'} / \tau_t\right)}, \tag{9}$$

where $d \in [D_k]$, $k \in [K]$ and $i \in [N]$. Since the parameter of the pre-trained teacher model is frozen, it is safe to assume that it will not produce the same representations for different instances $G_i, G_j \in \mathcal{G}$. So, we have:

$$\forall G_i, G_j \in \mathcal{G}, \quad \text{if } \mathbf{z}_{i,k}^t \neq \mathbf{z}_{j,k}^t \Longrightarrow \mathbf{z}_{i,k}^t \cdot \mathbf{q}_k \neq \mathbf{z}_{j,k}^t \cdot \mathbf{q}_k$$
$$\Longrightarrow \mathbf{p}_{i,k}^t \neq \mathbf{p}_{j,k}^t \Longrightarrow \mathbf{p}_{i,k}^t, \mathbf{p}_{j,k}^t \neq \frac{1}{D_k} \mathbf{1}_{D_k} \tag{10}$$

In this case, the sharpen operation will make the target probability distribution far more away from the uniform distribution. Then we have $\mathbf{p}_{i,k}^t \neq \mathbf{p}_{j,k}^s$ and $\|\nabla \mathcal{D}\left(\mathbf{p}_k^t, \mathbf{p}_k^s\right)\| > 0$ when $\mathbf{p}_{i,k}^s = \mathbf{p}_{j,k}^s = \frac{1}{D_k} \mathbf{1}_{D_k}$. Similarly, in the case of the $\mathbf{p}_{i,k}^s = \mathbf{p}_{j,k}^s \neq \frac{1}{D_k} \mathbf{1}_{D_k}$ for $G_i, G_j \in \mathcal{G}$, we have:

$$\bar{\mathbf{p}}_k^s = \frac{1}{N} \sum_{i=1}^{N} \mathbf{p}_{i,k}^s \neq \frac{1}{D_k} \mathbf{1}_{D_k}. \tag{11}$$

Therefore we can get $\|\nabla \mathcal{H}(\bar{\mathbf{p}}_k^s)\| > 0$ to force all the prototypes are utilized so that the instance with different semantics will not collapse into the same cluster (prototype).

# C  PROOF OF THEOREM 1

**THEOREM 1.** *For a graph dataset $\mathcal{G}$ and multiple student model $\{f_k^s\}_{k=1}^{K}$ with the different initialized knowledge, the ensemble of their predictions can reduce the error rate compared with the prediction*

*of any single model, i.e,*

$$\mathbb{E}_{\mathcal{G}}\left[\mathbf{1}\left(\hat{f}^s(G) = y\right)\right] \geq \mathbb{E}_{\mathcal{G}}\left[\mathbf{1}\left(f_k^s(G) = y\right)\right] \quad \forall k \in [K],$$

where $y$ is the ground truth label of instance $G$.

*Proof.* In the work, we train $K$ student models to extract features in different granularities, so each student model will utilize information from different views or levels to make predictions. Here, we denote the feature sets utilized by the $K$ student models as $\{\mathcal{M}_k\}_{k=1}^{K}$. For classification problems, the final prediction of the bagging ensemble can be obtained through majority voting:

$$\hat{f}^s(G) = \operatorname{argmax}\left(\Sigma\left(\mathbf{1}\left[f_k^s(G) = y\right]/K\right)\right). \tag{12}$$

To get the conclusion in Theorem 1, we start with the definition of expected prediction error:

$$EPE(G) = E\left[(y - \hat{f}(G))^2\right] = E\left[y^2 - 2y\hat{f}(G) - \hat{f}(G)^2\right]$$
$$= E\left[y^2\right] - E\left[2y\hat{f}(G)\right] - E\left[\hat{f}(G)^2\right]. \tag{13}$$

Given the equation above, we can decompose the expected prediction error into three parts. First of all, since we model $y = f + \epsilon$, then we have:

$$E\left[y^2\right] = E\left[(f^+\epsilon)^2\right] = E\left[f^2\right] + 2E[f^\epsilon] + E\left[\epsilon^2\right]$$
$$= f^2 + 2f^E[\epsilon] + E\left[\epsilon^2\right], \tag{14}$$

where $\epsilon$ has zero mean and variance $\sigma^2$, so we can reach to $E\left[y^2\right] = f^+\sigma^2$. Then, the second term in Equation 13 can be derived into:

$$E[y\hat{f}] = E[(f + \epsilon)\hat{f}] = E[f\hat{f}] + E[\epsilon\hat{f}]$$
$$= E[f\hat{f}] + E[\epsilon]E[\hat{f}] = fE[\hat{f}]. \tag{15}$$

Thirdly, based on the definition of the variable variance, we show that:

$$E\left[\hat{f}^2\right] = \operatorname{Var}(\hat{f}) + E[\hat{f}]^2. \tag{16}$$

Therefore, by combining Equation 14, 15 and 16 together, we can get prediction error of of classifier $\hat{f}$ as:

$$EPE(G) = E\left[(y - \hat{f}(G))^2\right] = f^2 + \sigma^2 - 2fE[\hat{f}] + \operatorname{Var}[\hat{f}] + E[\hat{f}]^2$$
$$= \underbrace{(f - E[\hat{f}])^2}_{\text{Bias}} + \underbrace{\operatorname{Var}[\hat{f}]}_{\text{Variance}} + \underbrace{\sigma^2}_{\text{Noise}}. \tag{17}$$

Our goal is to reduce the variance term, given the relation between of and expectation and variance, we can get the variance of the ensemble predictions based on Equation 12:

$$\operatorname{Var}(\hat{f}^s(G)) = E\left[(\hat{f}^s(G) - E[\hat{f}^s(G)])^2\right]$$
$$= E\left[\left(\operatorname{argmax}\left(\Sigma\left(\mathbf{1}\left[f_k^s(G) = y\right]/K\right)\right) - E\left[\operatorname{argmax}\left(\Sigma\left(\mathbf{1}\left[f_k^s(G) = y\right]/K\right)\right)]^2\right)\right] \tag{18}$$
$$= \operatorname{Var}(f_k^s(G))/K.$$

Given the assumption that each prediction task is a binary classification problem, we can set the $p$ as the probability that the ground truth label for graph instance $G$ and the $1 - p$ as the probability that the ground truth label is 1. Then we can reach the variance of the ensemble model by combining Equation 18 and Taylor series expansion:

$$\operatorname{Var}(f_k^s(G)) = p \times (1 - p)/K. \tag{19}$$

Therefore, the variance between the observation $\hat{f}^s$ and the optimal model $f^*$ based on the training set will be reduced when the number of the classifiers increases. Due to the reason that each student model $f_k^s$ is trained on the same training set, we have the $E\left[\hat{f}^s\right] = E\left[f_k^s\right]$ for any $k \in [K]$. Therefore, the bias term of the expected prediction error generated by each classifier is equal. In that case, the ensemble model $\hat{f}^s$ with a lower variance will produce lower prediction error:

$$E\left[(y - \hat{f}^s(G))^2\right] \leq E\left[(y - f_k^s(G))^2\right] \quad \forall k \in [K]. \tag{20}$$

Thus, the conclusion in Theorem 1 is proved.

# D  DATASET STATISTICS

Table 4: Statistics of MoleculeNet datasets and protein-protein interaction network datasets.

| Dataset | #Graphs | Avg #Nodes | Avg Degree | #Tasks (Class) | Metric | Category |
|---------|---------|-----------|-----------|---------------|--------|----------|
| ZINC15 | 2,000,000 | 26.62 | 57.72 | - | - | biochemical |
| PPI-306K | 306925 | 39.82 | 729.62 | - | - | Protein-Protein Intersection Networks |
| BBBP | 2,039 | 24.06 | 51.90 | 1 | ROC-AUC | biochemical |
| Tox21 | 7,813 | 18.57 | 38.58 | 12 | ROC-AUC | biochemical |
| ToxCast | 8,576 | 18.78 | 38.62 | 617 | ROC-AUC | biochemical |
| SIDER | 1,427 | 33.64 | 70.71 | 27 | ROC-AUC | biochemical |
| ClinTox | 1,477 | 26.15 | 55.76 | 2 | ROC-AUC | biochemical |
| MUV | 93,087 | 24.23 | 52.55 | 17 | ROC-AUC | biochemical |
| HIV | 41,127 | 25.51 | 54.93 | 1 | ROC-AUC | biochemical |
| BACE | 1,513 | 34.08 | 73.71 | 1 | ROC-AUC | biochemical |
| PPI | 88000 | 49.35 | 890.77 | 40 | ROC-AUC | Protein-Protein Intersection Networks |
| Cora | 1 | 2,708 | 5,429 | 7 | Accuracy | Citation Networks |
| Pubmed | 1 | 19,717 | 44,338 | 3 | Accuracy | Citation Networks |
| ogbn-arxiv | 1 | 169,343 | 1,166,243 | 40 | Accuracy | Citation Networks |

In the work, we use the data from the chemistry domain, biology domain, and academia domain for evaluation. For the graph-level tasks, we will do the pre-training and distillation on a large unlabeled dataset, correspondingly. Then, the model will be finetuned on eight chemistry datasets and one biology dataset for evaluation. The chemistry datasets are sampled from MoleculeNet (Wu et al., 2018) and the biology dataset comes form protein ego-networks. For the node-level tasks, we follow the SSL evaluation setting adopted in previous work (Hou et al., 2022; Thakoor et al., 2022) and evaluate our proposed method in three citation network datasets. The statistics of these datasets are shown in Table 4.

# E  IMPLEMENTATION DETAILS

**Architectures.** In the work, we follow the previous works to choose the model architectures in different tasks. For the graph-level tasks, we follow (You et al., 2020; Xu et al., 2021; Suresh et al., 2021) on this setting to use a 5-layer-GIN encoder (Xu et al., 2018) followed by a 2-layer MLP as the architecture of the teacher model and student model. Specifically, we set ReLU as the activation function and BathNorm as the normalization function. In the node-level tasks, we follow (Hou et al., 2022; Thakoor et al., 2022) to select a 2-layer GCN encoder as the student as the model architecture of student models and teacher models.

**Training Details.** All the teacher models are pre-trained based on the provided settings of the original authors. During the distillation process, the parameters of the teacher model are frozen and we randomly initialize $K$ student models with the same architecture as the teacher model. Meanwhile, K sets of the prototypes are also randomly initialized for the self-supervised prototypical contrastive distillation. We set $K = 3$ and the prototype number (i.e., $\{D_k\}_{k=1}^K$) to $\{50, 21, 2\}$. The student temperature hyper-parameter and the sharpening parameter are as $\tau_s = 0.1$ and $P = 0.25$ and we train the student models for 100 epochs. During the fine-tuning phase, we still follow the previous works to append a 1-layer MLP at the end of the GNN encoders for all student models and train them to adapt to the downstream tasks for 100 epochs. We use the Adma optimizer for the gradient descent of both knowledge distillation and fine-tuning, and the cosine decay learning rate scheduler is used in the knowledge distillation process. To guarantee the generalization of the framework, some commonly used graph augmentation techniques are used in this work, including edge perturbation, node dropping, and subgraph sampling.

**Key Hyper-parameters.** As we introduced, the experimental results in the overall performance table are conducted with ( student models (3 semantic levels), and their corresponding granularities are $\{2, 21, 50\}$ respectively. Meanwhile, more experiments about the performance sensitivity on student model (semantic level) numbers are also conducted and introduced in the experiment section. Another hyper-parameter $\lambda$ (ME-MAX regularization weight) introduced by our method is 1.0. For other important hyper-parameters in this work, we set the learning rate is fixed as 0.001, the weight decay is set as 0.0005, the dropout ratio as 0 during knowledge distillation and 0.5 in fin-tuning,

the warmup epoch as 10, the student temperature hyper-parameter as $\tau_s = 0.1$ and the sharpening parameter as $P = 0.25$.

# F  EXTRA EXPERIMENTAL RESULTS AND ANALYSIS

In this section, we present the experimental results on additional datasets that were not included in the paper due to page limitations. Besides that, more empirical analysis will be demonstrated to further prove the clarifications in the paper.

Table 5: Ablation study of the proposed MGSE with three teacher models on other four chemistry domain datasets and the biology domain dataset.

| Model | Cora | Pubmed | ogbn-arxiv |
|---|---|---|---|
| GraphCL | $81.74_{\pm0.45}$ | $79.22_{\pm0.32}$ | - |
| GraphCL+MGSE | $\mathbf{82.42}_{\pm0.38}$ | $\mathbf{81.04}_{\pm0.33}$ | - |
| GraphMAE | $82.42_{\pm0.64}$ | $80.78_{\pm0.68}$ | $70.55_{\pm0.16}$ |
| GraphMAE+MGSE | $\mathbf{83.12}_{\pm0.47}$ | $\mathbf{81.58}_{\pm0.35}$ | $\mathbf{71.68}_{\pm0.22}$ |
| BGRL | $82.80_{\pm0.53}$ | $80.52_{\pm0.18}$ | $71.20_{\pm0.21}$ |
| BGRL+GMSE | $\mathbf{83.13}_{\pm0.26}$ | $\mathbf{81.47}_{\pm0.40}$ | $\mathbf{72.21}_{\pm0.15}$ |

**Overall Performance Comparison in Node-level Task.** We conduct experiments on three node-level datasets, including Cora, PubMed, and Ogbn-Arxiv. We take GraphCL, GraphMAE, and BGRL as the teacher model and still use the same architecture as the teacher model (a two-layer GCN) for the distilled student models. The results are shown in Table 5, from the results above we can find our proposed MGSE can consistently boost the performance of the teacher model. Specifically, the improvements on the larger datasets are more obvious, which is aligned with our expectations since large datasets usually include more complex semantic patterns.

Table 6: Ablation study of the proposed MGSE with three teacher models on other four chemistry domain datasets and the biology domain dataset.

| Model | ClinTox | HIV | BACE | MUV | PPI |
|---|---|---|---|---|---|
| w/o. sharpen | $76.59_{\pm3.05}$ | $78.05_{\pm0.97}$ | $78.50_{\pm1.25}$ | $70.32_{\pm0.91}$ | $67.46_{\pm1.02}$ |
| w/o. ME-MAX | $75.75_{\pm2.70}$ | $78.40_{\pm0.99}$ | $77.75_{\pm1.10}$ | $70.10_{\pm1.22}$ | $67.20_{\pm0.86}$ |
| GraphCL+GMSE | $\mathbf{78.67}_{\pm2.89}$ | $\mathbf{79.07}_{\pm0.72}$ | $\mathbf{79.22}_{\pm0.93}$ | $\mathbf{71.46}_{\pm1.45}$ | $\mathbf{69.11}_{\pm0.70}$ |
| w/o. sharpen | $82.10_{\pm3.15}$ | $76.14_{\pm1.02}$ | $78.74_{\pm1.34}$ | $72.10_{\pm0.84}$ | $64.51_{\pm1.14}$ |
| w/o. ME-MAX | $81.45_{\pm3.44}$ | $76.35_{\pm1.25}$ | $78.15_{\pm0.95}$ | $72.43_{\pm1.15}$ | $64.02_{\pm1.20}$ |
| JOAO+GMSE | $\mathbf{83.30}_{\pm2.44}$ | $\mathbf{77.50}_{\pm0.67}$ | $\mathbf{79.82}_{\pm0.71}$ | $\mathbf{73.52}_{\pm0.62}$ | $\mathbf{65.37}_{\pm0.96}$ |
| w/o. sharpen | $82.15_{\pm2.33}$ | $77.10_{\pm1.34}$ | $81.74_{\pm1.30}$ | $75.90_{\pm0.87}$ | $67.11_{\pm1.12}$ |
| w/o. ME-MAX | $81.78_{\pm2.86}$ | $76.77_{\pm0.96}$ | $82.02_{\pm1.75}$ | $75.36_{\pm1.20}$ | $66.42_{\pm1.56}$ |
| GraphLoG+GMSE | $\mathbf{83.72}_{\pm2.02}$ | $\mathbf{78.64}_{\pm0.80}$ | $\mathbf{83.18}_{\pm1.24}$ | $\mathbf{77.22}_{\pm0.94}$ | $\mathbf{68.26}_{\pm1.06}$ |

**Ablation Study on Prototype Optimization.** First, we show the ablation study results on the other five datasets (including the other four chemistry domain datasets and the biology domain dataset) in Table 6, from which we can generally get similar observations with the results on other four datasets, i.e., MGSE can consistently outperform the two variants, and both of the components are significant in boosting the performance of teacher model. Therefore, it is fair to say our design can help the whole framework to extract multi-granularity semantic features and produce discriminative representations for each graph instance.

**Performance Analysis on Prototype Initialization Strategy.** The left subplot of Figure 6 is experimental results with different prototype initialization strategies on another four datasets, including ClinTox, BACE, HIV, and MUV. Still, the experimental results are quite similar to what is observed in the other four datasets. Thus, we believe that the newly initialized prototypes could improve the model generalization ability based on the teacher model and train the student models with higher potentials.

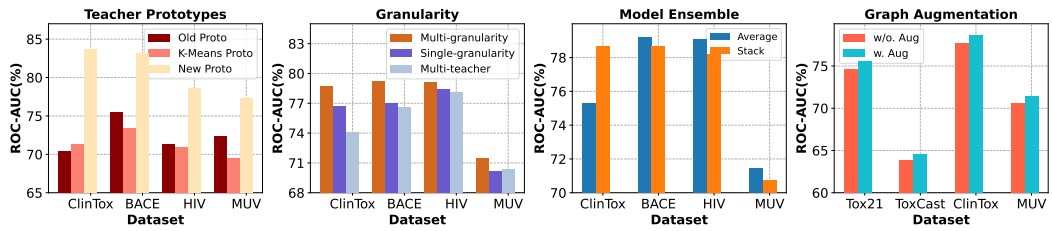

Figure 6: The impacts of different prototype initialization strategies (left), single-granularity versus multi-granularity (middle left), model ensemble strategies (middle right), and with versus without graph augmentation (right).

**Performance Analysis on Multi-granularity Semantic Features.** The middle left subplot presents the additional experimental results about single-granularity versus multi-granularity, which can further prove the conclusion we draw in the paper. Models with diversified semantics granularities can represent the graph in a more comprehensive manner, thereby resulting in a larger performance boost compared with the single-granularity variant. Moreover, the ensemble of teacher model outputs with different dropout ratios also yields sub-optimal performance compared with MGSE. We think the reason behind the phenomenon could be two-fold: (1) High dropout ratio may change the sample identity of each graph so that the generated representations contain insufficient information to make the correct prediction. On the other hand, a low dropout ratio could make the fine-tuned model overfitting on the trivial details and thereby have low generalization ability on the testing datasets, especially on the out-of-distribution setting (split by scaffold). Consequently, neither of them will produce positive effects on the final predictions; (2) The trained student models can benefit from the knowledge distillation.

**Performance Analysis on Model Ensemble Strategy.** The experimental results shown in the middle right subplot of Figure 6 are the additional results of the model ensemble strategy on another four datasets. As we can find in the figure, the experimental results are almost consistent with that of the other four datasets we demonstrate in the paper except for the ClinTox dataset. One possible explanation is that the classification task of ClinTox might need to consider the information across different granularity levels simultaneously. For example, some properties of a molecule are determined by a few functional groups (subgraph) but those functional groups belong to different granularities. In this case, a graph representation that contains information from different semantic levels could make better predictions.

**Performance Analysis on Graph Augmentation.** In the proposed MGSE, we incorporate graph augmentations in the framework because we believe augmentation is helpful and necessary to improve the model's generalization ability. Though equipped with a multi-granular knowledge distillation module, MGSE essentially is a self-supervised framework without external supervision signals. Hence, augmentation helps the model (either the student or teacher models) to learn representations invariant to perturbations, which aligns with the philosophy of all self-supervised-based frameworks. To empirically support our claim, we conduct additional experiments w.r.t. whether using graph augmentations or not over four datasets, the results are demonstrated in the right subplot of Figure 6, from which we observe that moderate augmentation will enhance the model performance.

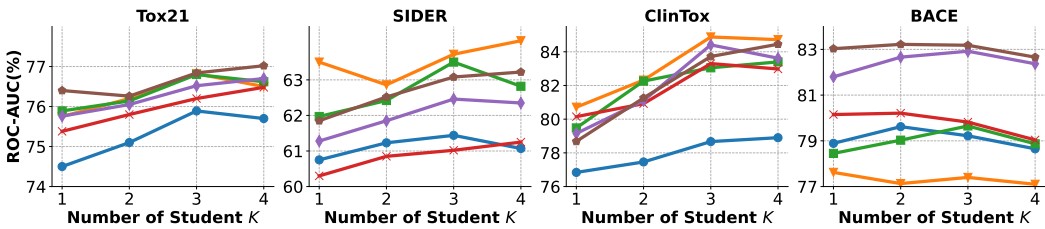

Figure 7: The impacts of different selections of the number of students over the other four chemistry domain datasets.

**Performance Analysis on Student Number.** Figure 7 above is experimental results with different numbers of students (the number of semantic granularities) on another four datase1ts, including Tox21, SIDER, ClinTox, BACE. Generally, the empirical observations on BBBP and ToxCast can be also applied to other datasets. For those multi-label datasets, i.e., Tox21, SIDER, and ClinTox, we can generally achieve better performance with a relatively large $K$. However, this advantage of multi-student models is less obvious in the single-label task, i.e., BACE. Once again, the phenomenon demonstrates that we need more comprehensive features from different views and perspectives (different semantic granularities) to handle various downstream applications, especially those multi-label classification problems.

# G    CASE STUDY

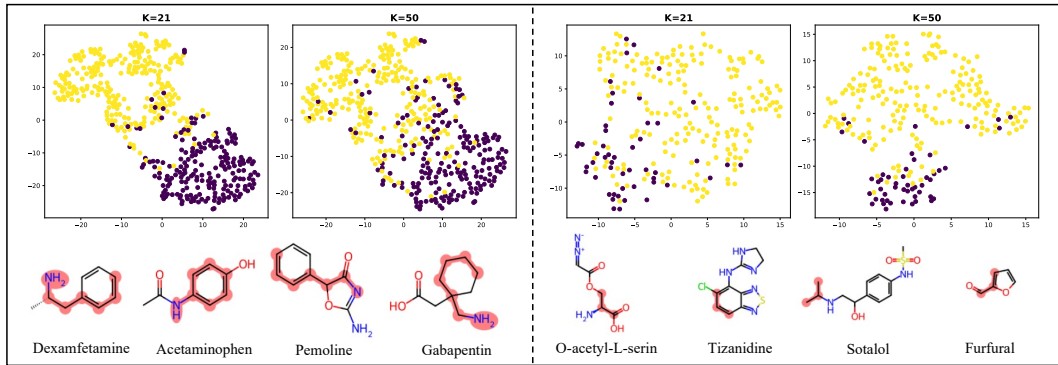

Figure 8: Embedding visualization results generated by student models with different granularities ($K = 21$ and $K = 50$) on BBBP (left) and Tox21 (right) datasets and the example molecules in the two datasets, the common substructures of molecules from the same dataset are highlighted in red. The four demonstrated example molecules of each dataset are all positive samples (labeled as 1) and all molecules sharing the same label as them in the dataset are represented in purple (otherwise in yellow) in the embedding visualization results.

In this section, we investigate the impact of knowledge in different granularities on diverse downstream tasks. To this end, we employ the t-SNE algorithm to visualize the embeddings generated by student models trained with varying granularities ($K = 21$ and $K = 50$). We use the BBBP dataset for a single-label task and the Tox21 dataset for a multi-label task as case studies to illustrate the differences. In Figure 8, the results show that the cluster-wise distance of visualization results from models trained with coarse-granularity ($K = 21$) is larger than those from fine-granularity ($K = 50$) on the BBBP dataset, which suggests that coarse-granularity information is more informative for classifying the BBBP dataset. Meanwhile, in the case of the Tox21 dataset, models trained with fine-granularity ($K = 50$) exhibit larger cluster-wise distances than those trained with coarse-granularity ($K = 21$), indicating fine-granularity information is more beneficial for the classification of the Tox21 dataset. To provide further insights into the represented information, we highlight four positive-labeled samples (purple) from each dataset (BBBP and Tox21) and identify their common substructures with red coloring. Despite lacking biological background, we can observe that the common substructure size of positive samples from the BBBP dataset is larger than that from the Tox21 dataset. This implies that the classification of BBBP relies on high-level abstract features, whereas fine-grained substructure information is more helpful to the classification of the Tox21 dataset.

# H    LIMITATION AND POTENTIAL IMPACT

**Limitation.** In this work, we focus on incorporating the multi-granularity semantic features to improve the model generalization ability on tasks like molecular and protein property predictions. Besides that, we can explore the effectiveness of the proposed MGSE in more domains, including but not limited to the analysis of social networks and brain networks. Moreover, the study about further decreasing the model architecture size with more advanced knowledge distillation techniques is also worthwhile for further exploration.

**Potential Impact.** As a plug-in framework to further increase the generalization ability of the current graph SSL methods across different downstream applications, we believe MGSE will not cause any societal negative impacts. On the other hand, MGSE can be helpful in the research on the explainability of the graph learning community. With the multi-granularity semantic features extracted by MGSE, we can provide a more precise and understandable depiction for each graph instance. Another promising direction could be the development of the fundamental model across different domains with the concept of multi-granularity features, where the 1-vs-K knowledge distillation framework can be extended into the N-vs-K form to further increase the capacity of the trained model.

