# OpenReview forum: "Graph Representation Learning with Multi-granular Semantic Ensemble"
_ICLR.cc/2024/Conference — Submitted to ICLR 2024_

### Official Review · Reviewer_YYfu · 2023-10-22

**Soundness:** 3 good
**Presentation:** 2 fair
**Contribution:** 2 fair
**Rating:** 6
**Confidence:** 4

**Summary:**

This paper proposes a teacher-multi-student knowledge distillation framework to enhance the effectiveness of existing graph SSL methods. Specifically, authors deem an existing SSL pre-trained graph encoder as teacher, and employ multiple students to acquire discriminative representations on different granularities guided by the teacher. On downstream tasks, the students are separately fine-tuned and their predictions are linearly combined as the final output. The proposed framework is shown to be able to improve six existing graph SSL algorithms on molecular graph and PPI graph modeling.

**Strengths:**

+ The proposed method is technically sound to extract discriminative representations in different semantic spaces, which shows some novelty and practical value.
+ The proposed framework is guaranteed with decent theoretical results.
+ The empirical results are sufficient to demonstrate the general effectiveness of the proposed framework.

**Weaknesses:**

- The effectiveness of the proposed framework is largely depended by the selection of prototype numbers, which determines the semantic levels learned by student models. However, the selection procedure of this set of hyperparameters is not clearly justified in the current draft.

**Questions:**

Generally, I am convinced by the proposed techniques, while I have some concerns on the selection of prototype structures:
1. **Selection of prototype number**: It seems that authors use some heuristic methods to determine the prototype numbers for multi-student distillation. Such heuristic method can hardly capture the intrinsic semantic structures underlying the pre-training dataset. By comparison, the prototype number determination scheme explored in [a] can better discover such structure in a learnable way. Can authors justify their prototype number selection scheme against such method with some theoretical or empirical results?
2. **Visualization of learned prototypes**: Authors are suggested to show the semantic prototypes learned for both molecular graphs and PPI graphs. Such visualization can help to understand the learning mechanism of the proposed method.


[a] Allen, Kelsey, et al. "Infinite mixture prototypes for few-shot learning." International conference on machine learning. PMLR, 2019.

---

> ### Author Response · Authors · 2023-11-18
> **Response to Reviewer YYfu**
>
> Thank you for your valuable feedback and suggestions. We sincerely appreciate your acknowledgment of the contributions we make in this paper (i.e., the practical impact, theoretical analysis as well as experimental performance). Our detailed responses to your concerns are listed as follows and we put the related weaknesses and questions together for a clear illustration. We truly hope you can increase the score if our reply addresses your concerns.
>
> ### W1: The effectiveness of the proposed framework is largely depended by the selection of prototype numbers, which determines the semantic levels learned by student models. However, the selection procedure of this set of hyperparameters is not clearly justified in the current draft.
>
> **Q1: Selection of prototype number: It seems that authors use some heuristic methods to determine the prototype numbers for multi-student distillation. Such heuristic method can hardly capture the intrinsic semantic structures underlying the pre-training dataset. By comparison, the prototype number determination scheme explored in [1] can better discover such structures in a learnable way. Can authors justify their prototype number selection scheme against such a method with some theoretical or empirical results?**
>
> > Thank you for your valuable suggestion. We introduce the selection of the $K$ sets prototypes number in page 5 and the default setting in the key hyper-parameters of Appendix E. To illustrate it again, we assign the number of prototypes in each set in descending order, i.e., $D_1 > D_2 > \cdots > D_K > 1$. We highlight the corresponding part with green in the updated paper version. Besides, we conduct experiments to evaluate the impacts of different prototype set numbers in the experiment section (left subplot of Figure 5).
> Upon reviewing the paper of IMP, we find that the known label space ($C$) and annotated support set are prioritized considerations for the Infinite Mixture Prototypes (IMP) in [1]. The learnable prototype numbers (clusters) are inferred based on a specified condition (e.g., $\operatorname{min}_{c} d_ic < \lambda$). It is crucial to note that the priority conditions required by IMP may not be provided in the context of self-supervised Learning (SSL) settings, rendering IMP unsuitable for SSL tasks. Consequently, a direct comparison between the two prototype number selection schemes may not be entirely fair. Our work aims to learn multi-granularity semantic features to adapt to various downstream tasks. In this regard, a SSL-based framework ensures that the extracted features are sufficiently generalizable for subsequent fine-tuning in downstream applications.
> Meanwhile, we acknowledge the potential of an adaptive prototype number selection schema tailored for SSL tasks as an intriguing research direction. Designing specific criteria to determine conditions for adjusting prototype numbers in SSL settings could offer valuable insights and warrant exploration in future research endeavors.
>
> [1] Allen, Kelsey, et al. "Infinite mixture prototypes for few-shot learning." International conference on machine learning. PMLR, 2019.
>
> ### Q2: Visualization of learned prototypes: Authors are suggested to show the semantic prototypes learned for both molecular graphs and PPI graphs. Such visualization can help to understand the learning mechanism of the proposed method.
>
> > Thanks for your suggestion. To address your concern, we add a case study to visually illustrate the knowledge learned by student models trained with different granularities, showcasing their effectiveness across different downstream tasks. Due to the constraints of the open review platform, we are unable to directly upload figures. Therefore, we update the case study in Appendix G after the main paper. We kindly invite you to refer to Appendix G for additional details and visualizations.
>
> **The content above is our response to your current review, please let us know if you have other questions and concerns and we would happily respond.**

---

> ### Author Response · Authors · 2023-11-22
> **Kind Reminder to Reviewer YYfu**
>
> Dear reviewer YYfu,
>
> Today is the last day of the reviewer-author discussion, we are still looking forward to the chance to address any remaining questions or concerns you may have. Please feel free to share them with us, and we are prepared to engage in further discussions. Thank you for your time and consideration.
>
> Best regards,
>
> MGSE authors

---

### Official Review · Reviewer_panV · 2023-10-31

**Soundness:** 2 fair
**Presentation:** 2 fair
**Contribution:** 2 fair
**Rating:** 5
**Confidence:** 4

**Summary:**

The paper introduces a novel method, MGSE, which aims to enhance the capabilities of self-supervised learning (SSL) in the graph learning domain. The authors address the challenge of existing graph SSL frameworks in capturing both high-level abstract features and fine-grained features simultaneously. By employing knowledge distillation, MGSE captures multi-granular knowledge using multiple student models learning from a single teacher model. Experimental results indicate that MGSE consistently improves the performance of several existing graph SSL frameworks.

**Strengths:**

The paper addresses a relevant challenge in the graph SSL domain.
Experimental results indicate the potential of MGSE to improve various existing graph SSL frameworks.
The authors provide a comprehensive discussion and analysis of their experimental results, showcasing a deep understanding and thorough examination of the outcomes.

**Weaknesses:**

The primary methodology of the paper appears to draw significant inspiration from the ProtoNCE loss, originally from the visual domain, as candidly acknowledged by the authors in “our optimization objective can be an analogy to the ProtoNCE loss to maximize the mutual information at cluster-level”. This brings forth concerns regarding the novelty and distinctiveness of the paper's central contribution. While adapting this approach to the context of GNNs is commendable, it seems that the crux of the innovation might be largely credited to the ProtoNCE loss itself.

The structure of the paper, particularly in Section 4, lacks comprehensive detail. With only two pages dedicated to the methodology, certain aspects remain unclear. For instance, the authors mention the construction of "K sets of corresponding trainable prototypes" but do not elucidate how these prototypes are updated. This omission leaves room for ambiguity and confusion.

The experimental results, while positive, do not offer a comprehensive comparison with other fine-tuning methods, which would have provided a clearer picture of MGSE's relative advantages.

**Questions:**

Could the authors clarify the update mechanism for the "K sets of corresponding trainable prototypes"?

How does the proposed MGSE method compare with other fine-tuning techniques when applied to one base model?

Given the analogy to the ProtoNCE loss, what are the unique challenges and considerations when applying this loss to the GNN domain?

The paper argues that multiple student models exhibit diverse levels of granularity. So why not choose the best model for the downstream task but use an ensemble? It is essential to consider the possibility that combining models with varying levels of granularity might introduce conflicting or inconsistent information, leading to a negative impact on the overall performance of the downstream task. A thorough analysis and experimentation on the ensemble's potential drawbacks and benefits are warranted to address this concern.

---

> ### Author Response · Authors · 2023-11-18
> **Response to Reviewer panV (1/2)**
>
> Thank you for your valuable feedback and suggestions. We sincerely appreciate your acknowledgment of the contributions we make in this paper (i.e., the practical impact, as well as the comprehensive empirical analysis). Our detailed responses to your concerns are listed as follows and we put the related weakness and questions together for clear illustration. We truly hope you can increase the score if our reply addresses your concerns.
>
> ### W1: The primary methodology of the paper appears to draw significant inspiration from the ProtoNCE loss, originally from the visual domain, as candidly acknowledged by the authors in “our optimization objective can be an analogy to the ProtoNCE loss to maximize the mutual information at cluster-level”. This brings forth concerns regarding the novelty and distinctiveness of the paper's central contribution. While adapting this approach to the context of GNNs is commendable, it seems that the crux of the innovation might be largely credited to the ProtoNCE loss itself.
>
> ### Q3: Given the analogy to the ProtoNCE loss, what are the unique challenges and considerations when applying this loss to the GNN domain?
>
> > Thanks for raising this question. As introduced in the introduction part, the primary motivations of our work are to analyze the generalizability of existing graph self-supervised methods and propose a framework to improve it. Through experiments conducted on multiple well-established datasets across diverse domains, we empirically demonstrate that the promising performance observed in existing methods on one or two datasets often fails to translate into robust generalization across all datasets. Given the sub-optimal generalization ability and the diversified structural indicators in different graph-related tasks, we posit the assumption that **Could multi-granularity graph semantic features further improve the generalization ability of learned representations in different downstream applications?** and propose the corresponding MGSE framework to enable the learning of multi-granularity semantic features.  Though the ProtoNCE loss is not our contribution, we believe that the philosophy of combining multi-granularity graph semantics is the pivotal factor leading to performance improvement, which can taken as the most important contribution of this work. Our assumption is also supported by the comprehensive empirical study in this work, MGSE achieves the best data generalization by comparing with six state-of-the-art competitive baselines with performance improvement of up to 9.2%, facilitated by our proposed multi-granular semantic ensemble that extracts comprehensive representations from input graphs.
> While algorithmic and architectural innovations are critical for driving progress in the field, works such as [1, 2, 3] that offer robust and insightful empirical investigations and studies of different aspects of performance (i.e., multi-granular graph representations for stronger generalization) are also essential for advancing our collective understanding of the existing research gaps. Therefore, we believe that our work could be a valuable contribution to the research community.
>
> [1] Knowledge distillation: A good teacher is patient and consistent In CVPR 2022
>
> [2] What Makes for Good Views for Contrastive Learning? In Neurips 2020
>
> [3] Is Homophily a Necessity for Graph Neural Networks? In ICLR 2022
>
>
> ### W2: The structure of the paper, particularly in Section 4, lacks comprehensive detail. With only two pages dedicated to the methodology, certain aspects remain unclear. For instance, the authors mention the construction of "K sets of corresponding trainable prototypes" but do not elucidate how these prototypes are updated. This omission leaves room for ambiguity and confusion.
>
> > Thanks for bringing up this question, we briefly illustrate the update mechanism of the K set prototypes at the lower part of page 5 and the upper part of page 6. To avoid confusion, We highlight this part in the updated paper file for reference. Here, we introduce the update mechanism in more detail for illustration. In our proposed MGSE, the teacher model is fixed and the K set trainable prototypes are optimized along with the K student models. Specifically, we compute the probability distributions ($\mathbf{p}^{t}$ and $\mathbf{p}^{s}$) of the teacher model and K student models on the K sets of prototypes as Equation (2), then measure their divergence with negative cross entropy as Equation (3).
> The measured divergence will be used to optimize the K sets of prototypes and student models to align $\mathbf{p}^{t}$ and $\mathbf{p}^{s}$ through the same optimizer.
> Meanwhile, as introduced on page 5, we add a ME-MAX regularization, i.e., Equation (4), to enforce as many prototypes to be utilized. The ME-MAX regularization along with the sharpen operation will optimize the K sets of prototypes in a meaningful and discriminative manner.

---

> ### Author Response · Authors · 2023-11-18
> **Response to Reviewer panV (2/2)**
>
> ### W3: The experimental results, while positive, do not offer a comprehensive comparison with other fine-tuning methods, which would have provided a clearer picture of MGSE's relative advantages.
>
> ### Q2: How does the proposed MGSE method compare with other fine-tuning techniques when applied to one base model?
>
> > To clarify, though our proposed MGSE is applied on the base graph SSL methods to diversify knowledge in different granularities, it does not require supervised information during the training phase. Therefore, our method should be categorized as a self-supervised learning framework like all the base methods. During the fine-tuning phase, we follow the same protocol, which is introduced in Appendix E, to finetune our method and all the teacher models with supervised information.
> To the best of our knowledge, there is currently no graph learning framework designed explicitly to leverage the knowledge of a graph self-supervised learning (SSL) model and further enhance its generalization ability in a self-supervised manner.  We consider this aspect to be a novelty and a noteworthy contribution of our proposed framework. We hope our work can offer a distinctive perspective and serve as inspiration for future research endeavors aimed at advancing the study of comprehensive and robust graph representation learning.
>
> ### W4: The paper argues that multiple student models exhibit diverse levels of granularity. So why not choose the best model for the downstream task but use an ensemble? It is essential to consider the possibility that combining models with varying levels of granularity might introduce conflicting or inconsistent information, leading to a negative impact on the overall performance of the downstream task. A thorough analysis and experimentation on the ensemble's potential drawbacks and benefits are warranted to address this concern.
>
> > Thanks for raising this question. We acknowledge that the utility of fine-grained versus coarse-grained knowledge may vary across specific downstream tasks, and the selection of the most informative knowledge granularity is task-dependent. However, a central motivation behind our paper is the development of a model that exhibits adaptability across a broad spectrum of downstream tasks. In situations where downstream tasks are not predefined, representations with multi-granularity semantics is more promising to achieve competitive performances across various tasks. This is supported by the observation that our proposed Multi-Granularity Semantic Embeddings (MGSE) consistently enhance the performances of all teacher models across diverse datasets and settings. Additionally, there are instances where a single application(dataset) requires knowledge at different granularities, particularly in multi-task scenarios. Our experimental observations further support this claim by the more substantial performance improvements on multi-label datasets, including Tox21, ToxCast, SIDER, ClinTox, and MUV.
> > To further address your concern, we conduct additional experiments w.r.t. the performance of each single student model and the results are shown in the table below. We can observe that each student outperforms the teacher model (GraphCL), showing the effectiveness brought by the knowledge distillation. However, coupled with our proposed multi-granular ensemble mechanism, MGSE achieves the best performance across all variants and baselines.
>
> |                           |      BBBP      |     Tox21      |    ToxCast     |     SIDER      |    ClinTox     |      HIV       |      BACE      |      MUV       |
> | :-----------------------: | :------------: | :------------: | :------------: | :------------: | :------------: | :------------: | :------------: | :------------: |
> | Teacher |   69.68±0.67   |   73.87±0.66   |   62.40±0.57   |   60.53±0.88   |   75.99±2.65   |   78.47±1.22   |   75.38±1.44   |   69.80±2.66   |
> | MGSE (Student #1, K=50) |   71.20±0.92   |   75.33±0.70   |   63.85±0.35   |   60.83±0.65   |   77.37±1.28   |   77.87±1.48   |   77.66±0.64   |   71.06±1.33   |
> | MGSE (Student #2, K=21) |   71.68±0.77   |   75.48±0.50   |   62.06±0.30   |   59.78±0.58   |   77.16±1.54   |   78.25±1.08   |   78.26±0.90   |   69.75±1.43   |
> | MGSE (Student #3, K=2) |   68.75±0.69   |   72.88±0.70   |   60.29±0.58   |   57.66±0.32   |   74.63±1.06   |   76.84±1.65   |   75.16±0.87   |   68.73±1.25   |
> |  MGSE (Average Ensemble 3 students)  | **72.26±0.65** | **75.89±0.33** | **64.57±0.34** | **61.44±0.68** | **78.67±2.89** | **79.07±0.72** | **79.22±0.93** | **71.46±1.45** |
>
> **The content above is our response to your current review, please let us know if you have other questions and concerns and we would happily respond.**

---

> ### Author Response · Authors · 2023-11-22
> **Kind Reminder to Reviewer panV**
>
> Dear reviewer panV,
>
> Today is the last day of the reviewer-author discussion, we are still looking forward to the chance to address any remaining questions or concerns you may have. Please feel free to share them with us, and we are prepared to engage in further discussions. Thank you for your time and consideration.
>
> Best regards,
>
> MGSE authors

---

### Official Review · Reviewer_LCB5 · 2023-11-01

**Soundness:** 3 good
**Presentation:** 3 good
**Contribution:** 2 fair
**Rating:** 5
**Confidence:** 3

**Summary:**

This work studies a problem overlooked by existing graph self-supervised models, i.e., how to simultaneously capture coarse-grained and fine-grained information for outstanding performance in various downstream tasks. To this end, the authors propose a plug-and-play graph knowledge distillation framework (MGSE), which can integrate with existing graph self-supervised learning models and enhance model performance by incorporating multi-granularity concepts. Specifically, under the condition of probability distributions at different granularities, MGSE captures multi-granularity knowledge by making multiple student models learn from a single teacher model. Extensive results on several benchmarks demonstrate that the proposed MGSE improves the performance of existing graph self-supervised learning models. The ablation experiments also demonstrated the effectiveness of the techniques employed in MGSE.

In summary, this work makes the following contributions: Firstly, it proposes a plug-and-play knowledge distillation framework to enhance the generalization of any graph-based self-supervised learning model. Based on empirical experimental results, the framework demonstrates promising performance, and the authors provide theoretical guarantees for the performance of MGSE.

**Strengths:**

1. The considered problem is important, and the proposed method is technically sound.
2. Experiments conducted show that the proposed method achieves good empirical performance.
3. The paper is well-written and easy to follow.

**Weaknesses:**

While overall this work does not have major flaws, I still have some concerns as follows and I hope the authors to address them as much as possible.
1. Although the authors have designed different student models based on the analysis to capture knowledge at different granularities for solving various downstream tasks, it is important to visually demonstrate the distinct granularities of knowledge captured by each student model, rather than solely asserting that “different prototype sets capture different semantic granularities because we assign different numbers of prototypes to each prototype set in descending order”. The authors should consider adding a case study to provide a better explanation, like the mentioned example in the introduction about which knowledge in amino acids is coarse-grained and which knowledge is fine-grained.

2. In this paper, the framework captures multi-granular knowledge and performs an averaging operation on the outputs of different student models. However, since different datasets may lean towards different granularities of knowledge, it is worth considering whether the introduction of fine-grained knowledge in datasets where coarse-grained knowledge dominates could potentially be not only irrelevant to the target task but also introduce noise. Taking document classification as an example, capturing high-level textual features may be sufficient, and the fine-grained semantic features may not provide substantial assistance in determining the category of the document.

3. If I haven't missed any important details, as far as I know, the model ensemble can also lead to performance improvement. Therefore, it is crucial to determine whether the performance improvement comes from capturing knowledge at different granularities or is simply a result of the model ensemble strategy.

4. From the results, it can be observed that the designed framework brings some improvement, but this improvement comes at the cost of ensemble of multiple models. Additionally, although the authors have indicated that the proposed framework's computational complexity is proportional to the existing graph self-supervised models, considering the significant computational complexity of the original graph-based self-supervised models, it is uncertain whether this trade-off is worthwhile.

**Questions:**

Please see the weaknesses.

---

> ### Author Response · Authors · 2023-11-18
> **Response to Reviewer LCB5 (1/2)**
>
> Thank you for your valuable feedback and suggestions. We sincerely appreciate your acknowledgment of the contributions we make in this paper (i.e., the practical impact, experimental performance as well as writing clarity). Our detailed responses to your concerns are listed as follows. We truly hope you can increase the score if our reply addresses your concerns.
>
> ### W1: Although the authors have designed different student models based on the analysis to capture knowledge at different granularities for solving various downstream tasks, it is important to visually demonstrate the distinct granularities of knowledge captured by each student model, rather than solely asserting that “different prototype sets capture different semantic granularities because we assign different numbers of prototypes to each prototype set in descending order”. The authors should consider adding a case study to provide a better explanation, like the mentioned example in the introduction about which knowledge in amino acids is coarse-grained and which knowledge is fine-grained.
>
> > Thanks for your suggestion. To address your concern, we add a case study to visually illustrate the knowledge learned by student models trained with different granularities, showcasing their effectiveness across different downstream tasks. Due to the constraints of the open review platform, we are unable to directly upload figures. Therefore, we update the case study in Appendix G after the main paper. We kindly invite you to refer to Appendix G for additional details and visualizations.
>
>
> ### W2: In this paper, the framework captures multi-granular knowledge and performs an averaging operation on the outputs of different student models. However, since different datasets may lean towards different granularities of knowledge, it is worth considering whether the introduction of fine-grained knowledge in datasets where coarse-grained knowledge dominates could potentially be not only irrelevant to the target task but also introduce noise. Taking document classification as an example, capturing high-level textual features may be sufficient, and the fine-grained semantic features may not provide substantial assistance in determining the category of the document.
> > Thanks for raising this question. We acknowledge that the utility of fine-grained versus coarse-grained knowledge may vary across specific downstream tasks, and the selection of the most informative knowledge granularity is task-dependent. However, a central motivation behind our paper is the development of a model that exhibits adaptability across a broad spectrum of downstream tasks. In situations where downstream tasks are not predefined, representations with multi-granularity semantics is more promising to achieve competitive performances across various tasks. This is supported by the observation that our proposed Multi-Granularity Semantic Embeddings (MGSE) consistently enhances the performances of all teacher models across diverse datasets and settings. Additionally, there are instances where a single application(dataset) requires knowledge at different granularities, particularly in multi-task scenarios. Our experimental observations further support this claim by the more substantial performance improvements on multi-label datasets, including Tox21, ToxCast, SIDER, ClinTox, and MUV.
> To further address your concern, we conduct additional experiments w.r.t. the performance of each single student model and the results are shown in the table below. We can observe that each student outperforms the teacher model (GraphCL), showing the effectiveness brought by the knowledge distillation. However, coupled with our proposed multi-granular ensemble mechanism, MGSE achieves the best performance across all variants and baselines.
>
> |                           |      BBBP      |     Tox21      |    ToxCast     |     SIDER      |    ClinTox     |      HIV       |      BACE      |      MUV       |
> | :-----------------------: | :------------: | :------------: | :------------: | :------------: | :------------: | :------------: | :------------: | :------------: |
> | Teacher |   69.68±0.67   |   73.87±0.66   |   62.40±0.57   |   60.53±0.88   |   75.99±2.65   |   78.47±1.22   |   75.38±1.44   |   69.80±2.66   |
> | MGSE (Student #1, K=50) |   71.20±0.92   |   75.33±0.70   |   63.85±0.35   |   60.83±0.65   |   77.37±1.28   |   77.87±1.48   |   77.66±0.64   |   71.06±1.33   |
> | MGSE (Student #2, K=21) |   71.68±0.77   |   75.48±0.50   |   62.06±0.30   |   59.78±0.58   |   77.16±1.54   |   78.25±1.08   |   78.26±0.90   |   69.75±1.43   |
> | MGSE (Student #3, K=2) |   68.75±0.69   |   72.88±0.70   |   60.29±0.58   |   57.66±0.32   |   74.63±1.06   |   76.84±1.65   |   75.16±0.87   |   68.73±1.25   |
> |  MGSE (Average Ensemble 3 students)  | **72.26±0.65** | **75.89±0.33** | **64.57±0.34** | **61.44±0.68** | **78.67±2.89** | **79.07±0.72** | **79.22±0.93** | **71.46±1.45** |

---

> ### Author Response · Authors · 2023-11-18
> **Response to Reviewer LCB5 (2/2)**
>
> ### W3: If I haven't missed any important details, as far as I know, the model ensemble can also lead to performance improvement. Therefore, it is crucial to determine whether the performance improvement comes from capturing knowledge at different granularities or is simply a result of the model ensemble strategy.
> > Yes, we have conducted experiments to verify that the performance improvements are not solely gained from the model ensemble and we demonstrate experimental results and corresponding analysis in the right subplot of Figure 3 and page 8.
> In the experiments, we add two other model ensemble variants without multi-granularity semantics learning design in our proposed framework. From the results we can find, that though the two variants based on model ensemble can also boost the teacher model’s performance to some extent, the improvement over MGSE is obviously larger, indicating the effectiveness of considering multiple granularities other than the model ensemble. Meanwhile, it is fair to say that a simple model ensemble strategy can not bring performance improvements as significant as our multi-granularity semantics learning design and we propose a more effective method to learn comprehensive graph knowledge based on the model ensemble.
>
> ### W4: From the results, it can be observed that the designed framework brings some improvement, but this improvement comes at the cost of the ensemble of multiple models. Additionally, although the authors have indicated that the proposed framework's computational complexity is proportional to the existing graph self-supervised models, considering the significant computational complexity of the original graph-based self-supervised models, it is uncertain whether this trade-off is worthwhile.**
>
> > Thanks for bringing up this question. We admit that learning multi-granularity semantics through our designed framework incurs additional computational costs. However, we think that the trade-off is worthwhile for several reasons.
> > Firstly, GNNs are lightweight models compared to many backbones in other domains such as text and image processing. The computational complexity of our designs is proportional to existing graph self-supervised models, and thus, the scalability bottleneck of current graph SSL methods does not lie in the model size. Consequently, the performance improvements brought by our method do not significantly compromise computational efficiency.
> > Secondly, although not the primary focus of this work, our proposed Multi-Granularity Semantic Embeddings (MGSE) demonstrate the potential for model compression. We have conducted experiments to investigate the impact of GNN model depth on final performance, as illustrated in the right subplot of Figure 4. The experimental results indicate that a larger model architecture consistently contributes positively to the final performance. Moreover, our framework has the capacity to approximate or even surpass the performance of over-parameterized teacher models with multiple lighter student models. Consequently, the results illustrate that the additional computation cost incurred by incorporating multiple student models can be mitigated by achieving comparable results with a lighter student model architecture.
> > Finally, we respectively argue that a model architecture with sufficient capacity is paramount for enhancing generalization ability. Though our proposed MGSE can not achieve simialr effects to the foundational models in the NLP and CV domains, our goal is to improve the generalization ability of existing graph SSL methods. This enables them to seamlessly adapt to diverse downstream graph-related applications with more competitive performances, which is validated by the comprehensive experiments presented in our paper. Therefore, we believe the increased computation cost is justified in light of the broader goal of advancing model generalization ability.
>
> **The content above is our response to your current review, please let us know if you have other questions and concerns and we would happily respond.**

---

> ### Author Response · Authors · 2023-11-22
> **Kind Reminder to Reviewer LCB5**
>
> Dear reviewer LCB5,
>
> Today is the last day of the reviewer-author discussion, we are still looking forward to the chance to address any remaining questions or concerns you may have. Please feel free to share them with us, and we are prepared to engage in further discussions. Thank you for your time and consideration.
>
> Best regards,
>
> MGSE authors

---

### Author Response · Authors · 2023-11-21
**General Response to All Reviewers**

We sincerely appreciate the reviewers for their efforts and constructive feedback. We have made several updates to our manuscript to highlight a few details that can be omitted to avoid confusion. Meanwhile, we provide detailed responses to all the concerns and questions proposed by each reviewer. Specifically, we want to emphasize two key issues:

**1. Why not select one best model for each downstream task?**
As introduced in our paper, the primary motivation of this paper is to ensure that the trained model exhibits adaptability across as many downstream tasks as possible. While we acknowledge the possibility of selecting the best-performing student model and fine-tuning it for a specific downstream task, it is essential to note that this prior knowledge is often unavailable in the self-supervised setting where supervised information is not provided. To maintain the generalization ability of our framework, we adopt an average ensemble approach, as detailed in our paper. The effectiveness of this design choice is demonstrated by the consistent performance improvements achieved by MGSE across various datasets and settings. Additionally, there could be cases in which a single task needs knowledge in different granularities to make an accurate prediction. To verify it, we conduct additional experiments to evaluate each of student model. This is supported by the reported results in our responses to reviewers, where the averaged representation of student models can consistently outperform each individual student model.

**2. The visualization of learned knowledge.**
We also add visualization results to i provide an intuitive depiction of the acquired graph knowledge at different granularities.
To do so, we visualize the graph embeddings generated by two different student models (K=21 and K=50) and select BBBP (single-label) and Tox21 (multi-label) for illustration to investigate the effectiveness of different granularities in various tasks. Referencing the visualization results in **Appendix G**, we observe that coarse-granularity (K=21) knowledge is more effective for the BBBP dataset, while fine-granularity (K=50) proves more informative for Tox21, as evidenced by the larger cluster-wide distance.
Furthermore, we highlight common substructures of positive examples from the two datasets to dive into the reasons behind this phenomenon. The visualization of molecule graphs shows that the size of the common substructure for positive molecules in BBBP is notably larger than that in Tox21, suggesting that molecules of the same class in BBBP tend to share very similar patterns. Consequently, we believe that this distinction in scale indicates that more high-level abstract features (coarse-granularity) are more effective for classifying BBBP, whereas the model needs to discern molecules in Tox21 based on relatively small aspects (fine-granularity).

The important contributions of our work include the thorough analysis of the sub-optimal generalization ability observed in current graph self-supervised learning methods and we thereby propose the assumption **Could multi-granularity graph semantic features further improve the generalization ability of learned representations in different downstream applications?** Based on the assumption, we correspondingly design our MGSE framework to enable the learning of multi-granularity semantic features. Furthermore, we conduct comprehensive empirical and theoretical analysis to demonstrate the effectiveness of our proposed MGSE, which achieves the best data generalization by comparing with six state-of-the-art competitive baselines with performance improvement of up to 9.2%. While algorithmic and architectural innovations are critical for driving progress in the field, works such as [1, 2, 3] that offer robust and insightful empirical investigations and studies of different aspects of performance are also essential for advancing our collective understanding of the existing research gaps. Therefore, we believe that our work could be a valuable contribution to the research community.

**As the author-reviewer discussion nears its ending, we hope you can consider increasing your ratings if our responses have addressed your questions. If you have any remaining concerns, please do not hesitate to let us know, and we will be more than happy to provide further clarification. Thanks again!**

[1] Knowledge distillation: A good teacher is patient and consistent In CVPR 2022

[2] What Makes for Good Views for Contrastive Learning? In Neurips 2020

[3] Is Homophily a Necessity for Graph Neural Networks? In ICLR 2022

Best regards,

MGSE authors

---

### Meta-Review · Area_Chair_6h7G · 2023-12-06

**Metareview:**

The paper presents an approach for representation learning in graphs at different level of granularity, eventually using an average of models trained at different granularities.

The reviewers agree that the approach proposed in the paper make sense, but overall the reviewer support for acceptance remained lukewarm. Some clarification questions have been resolved by the rebuttal (e.g., adding new visualization), but no reviewer volunteered to champion the paper. Some outstanding questions remained regarding hyperparameter optimization (selecting the number of prototypes) or naive ensembling (why not select/reweight different granularities depending on the downstream task at hand).

Overall the paper is borderline, very close to the bar. The reviewers agree that the paper addresses an important problem, and that the experimental results are promising. However, the methodology is sound but without clear algorithmic or technical contribution, so my inclination is to recommend reject.

**Justification For Why Not Higher Score:**

A clearer technical contribution or more in-depth studies of what can be done with multi-granular representations would likely have raised the score.

**Justification For Why Not Lower Score:**

N/A

---

### Decision · Program_Chairs · 2024-01-16

Reject